# Effect of β2-agonist treatment on insulin-stimulated peripheral glucose disposal in healthy men in a randomised placebo-controlled trial

β2-agonist treatment improves skeletal muscle glucose uptake and whole-body glucose homeostasis in rodents, likely via mTORC2-mediated signalling. However, human data on this topic is virtually absent. We here investigate the effects of two-weeks treatment with the β2-agonist clenbuterol (40 μg/day) on glucose control as well as energy- and substrate metabolism in healthy young men (age: 18-30 years, BMI: 20-25 kg/m²) in a randomised, placebo-controlled, double-blinded, cross-over study (ClinicalTrials.gov-identifier: NCT03800290). Randomisation occurred by controlled randomisation and the final allocation sequence was seven (period 1: clenbuterol, period 2: placebo) to four (period 1: placebo, period 2: clenbuterol). The primary and secondary outcome were peripheral insulin-stimulated glucose disposal and skeletal muscle GLUT4 translocation, respectively. Primary analyses were performed on eleven participants. No serious adverse events were reported. The study was performed at Maastricht University, Maastricht, The Netherlands, between August 2019 and April 2021. Clenbuterol treatment improved peripheral insulin-stimulated glucose disposal by 13% (46.6 ± 3.5 versus 41.2 ± 2.7 μmol/kg/min, p = 0.032), whereas skeletal muscle GLUT4 translocation assessed in overnight fasted muscle biopsies remained unaffected. These results highlight the potential of β2-agonist treatment in improving skeletal muscle glucose uptake and underscore the therapeutic value of this pathway for the treatment of type 2 diabetes. However, given the well-known (cardiovascular) side-effects of systemic β2-agonist treatment, further exploration on the underlying mechanisms is needed to identify viable therapeutic targets.

Reduced skeletal muscle glucose uptake in response to insulin is central to the development of type 2 diabetes mellitus (T2DM)[1]. However, antidiabetic drugs aimed at improving skeletal muscle glucose uptake in T2DM patients are currently non-existent, underscoring the need for novel targets. In this context, β2-adrenergic receptor (β2-AR) agonists were shown to have profound effects on skeletal muscle glucose uptake in preclinical models. Indeed, stimulation of the β2-AR−the most abundant type of β-AR in skeletal muscle[2]−with either the selective β2-AR agonists zinterol or clenbuterol increased glucose uptake in L6 muscle cells[3−6]. In line with these findings, in vivo glucose uptake in gastrocnemius muscle was enhanced by ~74% in C56Bl/6 N mice upon 6 days of treatment with clenbuterol[6]. More prolonged

✉ e-mail: J.Hoeks@maastrichtuniversity.nl

treatment (4 days up to 5 weeks) with either low- or high doses of clenbuterol also improved both glucose and insulin tolerance in various rodent models of insulin resistance and type 2 diabetes[5,7–11].

Despite the mounting evidence that β₂-AR agonists improve skeletal muscle glucose uptake and whole-body glucose homeostasis, the exact underlying (molecular) mechanisms remain elusive. However, given the well-known (cardiovascular) side effects of systemic β₂-AR agonists, this information is essential to uncover novel potential targets for the treatment of insulin resistance and T2DM. In this context, previous in vitro data by Sato et al.[5] has revealed a role for the mammalian target of rapamycin complex 2 (mTORC2) in β₂-AR agonist-stimulated skeletal muscle glucose uptake, an effect that was mediated via glucose transporter 4 (GLUT4) translocation to the myocyte membrane. In accordance with these findings, it was recently demonstrated in an elegant series of experiments that the antidiabetic effects of clenbuterol appeared largely dependent on mTORC2, as improvements in glucose homeostasis upon clenbuterol treatment were reduced in mice ablated of skeletal muscle-specific Rictor—a key component of mTORC2[11].

Given these promising preclinical findings, we here investigated in a double-blinded, randomized, placebo-controlled, cross-over study whether clenbuterol treatment ($2 \times 20\,\mu g$/day for 14 days) could improve insulin-stimulated skeletal muscle glucose disposal in healthy, lean males, as assessed by the gold standard, two-step hyper-insulinemic-euglycemic clamp technique. To assess putative underlying physiological and molecular mechanisms, we performed in-depth metabolic phenotyping after clenbuterol treatment using state-of-the-art techniques. Thus, we determined body composition, assessed (sleeping) energy metabolism by whole-room calorimetry, femoral artery blood flow velocity using Doppler ultrasonography whereas muscle biopsies were taken in the overnight fasted state to test if clenbuterol treatment was associated with enhanced skeletal muscle mTORC2 activation and GLUT4 translocation.

Here, we show that clenbuterol treatment increases insulin-stimulated peripheral glucose uptake and non-oxidative glucose disposal (NOGD) in healthy young males by ~13 and ~18%, respectively. However, these effects occurred independent of changes in skeletal muscle GLUT4 translocation or mTORC2 activation. These results highlight the potential therapeutic value of targeting the β₂-adrenergic receptor to improve skeletal muscle glucose uptake, although the underlying mechanisms remain elusive.

## Results

### Participant characteristics

In total, eleven healthy young male participants (age: $24.9 \pm 3.7$ years, BMI: $23.2 \pm 1.8\,kg/m^2$) received a two-week treatment with clenbuterol ($20\,\mu g$ twice daily) versus placebo in a randomised, double-blinded, cross-over study design. Baseline participant characteristics are displayed in Table 1. Due to several drop-outs between inclusion of participants and the start of the intervention period, the final allocation ratio was seven (period 1: clenbuterol, period 2: placebo) to four (period 1: placebo, period 2: clenbuterol) (see Fig. 1). Compliance—measured as the number of capsules returned by participants divided by the total capsules dispensed—was >95% for both treatment periods. Side-effects were reported in five out of eleven participants and included tremors in the hands ($n = 5$), muscle ache, tense muscles or muscle cramps ($n = 4$), feeling anxious and/or restless ($n = 2$) and headache ($n = 2$). All side-effects disappeared upon withdrawal of the drug. No serious adverse events were reported during the study.

### Insulin-stimulated glucose uptake in peripheral tissues is augmented by clenbuterol

The primary outcome of the study, peripheral insulin-stimulated glucose disposal, was assessed by a two-step hyperinsulinemic-

**Table 1 | Participant characteristics at screening**

| Participant characteristics | Mean + SD |
|---|---|
| Participants (n) | 11 |
| Age (years) | 24.9 (3.7) |
| Height (cm) | 176.3 (4.4) |
| Weight (kg) | 72.2 (6.6) |
| BMI (kg/m²) | 23.2 (1.8) |
| *Blood pressure* | |
| Systolic pressure (mmHg) | 120 (9.9) |
| Diastolic pressure (mmHg) | 72 (6.7) |
| Heart rate (beats/min) | 58 (6.4) |
| *Blood parameters* | |
| ALAT (U/L) | 25 (11) |
| ASAT (U/L) | 23 (7) |
| γ-GT (U/L) | 19 (6) |
| Creatinine (µmol/L) | 82 (7) |
| eGRF CKD-EPI (mL/min/1.73 m) | 112.2 (9.1) |
| Haemoglobin (mmol/L) | 9.6 (0.5) |
| Potassium (mmol/L) | 4.25 (0.22) |
| TSH (mU/L) | 2.2 (1.0) |

Data are presented as mean (standard deviation) for $n = 11$. *BMI* body mass index, *ALAT* alanine transaminase, *ASAT* aspartate aminotransferase, *γ-GT* gamma-glutamyl transferase, *eGRF* estimated glomerular filtration rate, *TSH* thyroid-stimulating hormone. Source data are provided as a Source Data file.

euglycemic clamp using stable isotope-labeled glucose performed after 14 days of clenbuterol treatment vs. placebo. The change in insulin-stimulated glucose disposal in the high insulin phase over baseline (ΔRd) was -13% higher upon clenbuterol treatment ($p = 0.032$, Fig. 2A), indicating an augmented glucose uptake into the peripheral tissues, primarily skeletal muscle. This occurred independent of changes in baseline rate of glucose disposal (Rd) or endogenous glucose production ($p = 0.320$ and $p = 0.765$, respectively, Fig. 2B, C). Similarly, hepatic insulin sensitivity—expressed as the percentage EGP suppression during the low-insulin phase of the clamp—was unaffected by clenbuterol treatment ($p = 0.700$, Fig. 2D). Additionally, the percentual suppression of plasma FFA during the low- and high-insulin phases of the clamp, which reflects adipocyte insulin sensitivity, remained unaffected upon clenbuterol treatment (Supplementary Fig. 1). The increased insulin-stimulated glucose disposal (ΔRd) was primarily accounted for by a -18% higher ΔNOGD ($p = 0.054$, Fig. 2E, F), whereas carbohydrate oxidation during the high-insulin phase of the clamp remained unaffected ($p = 0.592$, Supplementary Fig. 2B).

### No GLUT4 translocation or mTORC2 activation upon clenbuterol treatment

To gain further insights into the putative underlying molecular mechanisms through which clenbuterol stimulates peripheral glucose uptake, skeletal muscle biopsies were taken in the overnight fasted state after 14 days of clenbuterol treatment and were examined for GLUT4 translocation, mTORC2 activation, and markers of mitochondrial capacity. The secondary outcome of the study was defined as the fraction of total GLUT4 detected at the cellular membrane, reflecting GLUT4 translocation. Both GLUT4 translocation and total GLUT4 content remained unaffected by clenbuterol treatment ($p = 0.820$ and $p = 0.734$, respectively, Fig. 3A–D). In addition, we were unable to detect differences in the phosphorylation of mTOR S2481, a marker for mTORC2 activation, between clenbuterol and placebo ($p = 0.416$, Fig. 3E). Similarly, no changes were observed in the protein content of structural components of the different OxPhos complexes, or in the outer membrane proteins VDAC and TOMM20, all markers of mitochondrial capacity (Supplementary Fig. 3).

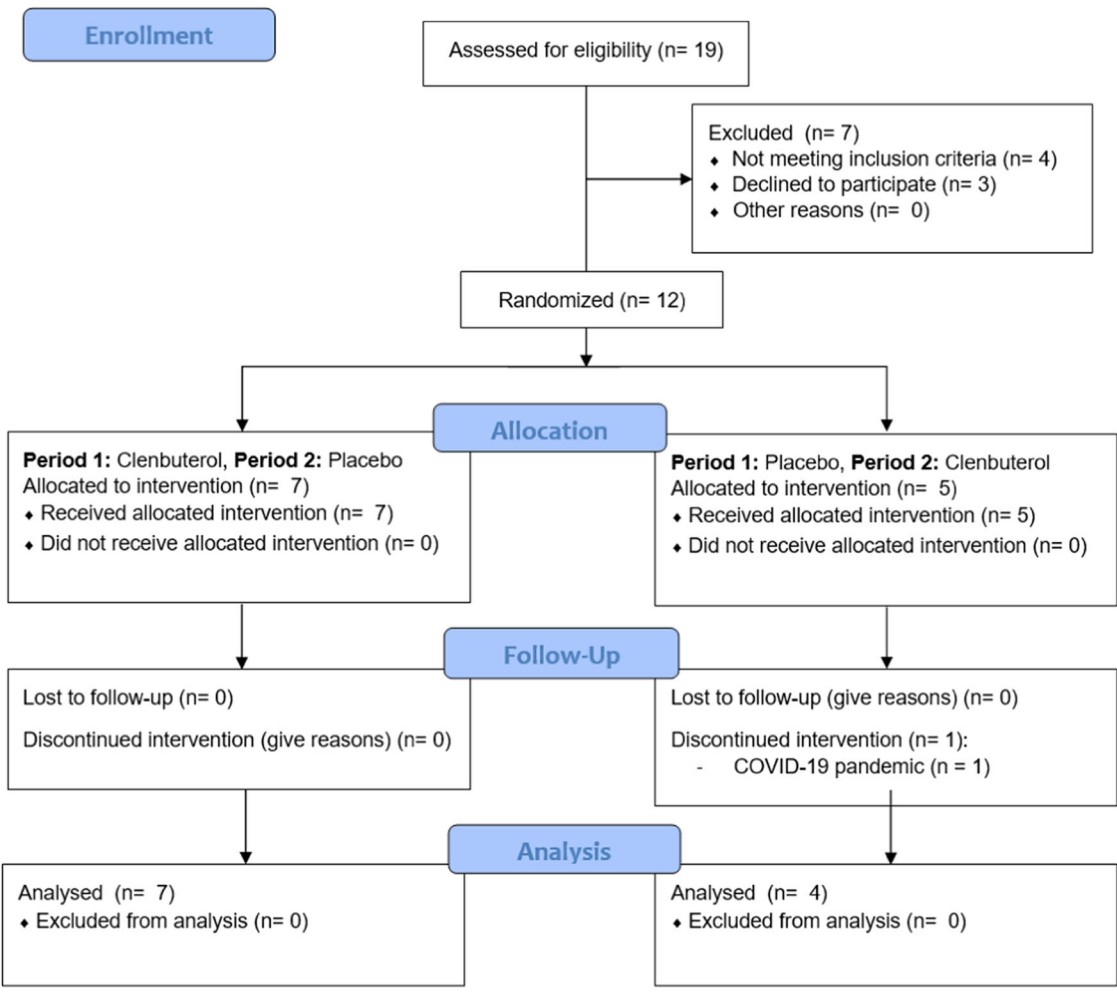

**Fig. 1 | CONSORT diagram inclusion flow-chart of the study.** P1 period 1, P2 period 2.

## Clenbuterol treatment increases femoral artery blood flow velocity

As we were unable to ascribe the effects of clenbuterol treatment to an enhanced skeletal muscle GLUT4 translocation or mTORC2 activation, we next investigated potential physiological mechanisms underlying the enhanced insulin-stimulated peripheral glucose uptake and first focussed on haemodynamic outcome parameters. Thus, two weeks of clenbuterol administration tended to significantly increase systolic blood pressure by ~3 mmHg ($p = 0.096$, Fig. 4A), whereas it increased heart rate by ~11 beats/min ($p = 0.003$, Fig. 4C). Despite these changes, both systolic blood pressure and heart rate remained within normal clinical values. Next, we determined blood flow responses and endothelial function using flow-mediated vasodilation (FMD) measurements of the femoral artery. Baseline femoral blood flow velocity was 16.7% higher upon clenbuterol treatment as compared to placebo ($p = 0.014$, Fig. 4D), whereas baseline femoral artery diameters were significantly increased by ~6% ($p = 0.005$, Fig. 4E). However, we did not detect any changes in endothelial function upon clenbuterol treatment, i.e., the FMD responses were not affected (Fig. 4F, G).

## The effect of clenbuterol on body composition and protein metabolism

As high doses of clenbuterol are well-known for their repartitioning effects on body composition (i.e., increasing lean mass and reducing fat mass)[7–9,12] and changes in protein metabolism[11,13], we investigated the effects of 2-weeks clenbuterol treatment on body weight and -composition. Body weight remained unaffected

($p = 0.915$, Supplementary Fig. 4A) and no effects on body composition were observed, as both fat- and lean mass were similar between the clenbuterol and placebo group (Supplementary Fig. 4B–E). Since two weeks of clenbuterol treatment may be too short to expect changes in net protein balance and body composition, we also measured the phosphorylation of mTOR S2448 (mTORC1), a marker of protein synthesis, as well as plasma amino acid levels. In line with the non-significant effects on lean mass, mTORC1 activation was unaffected upon clenbuterol treatment (Supplementary Fig. 5A). However, 2-weeks of clenbuterol treatment did significantly reduce fasting plasma concentrations of 12 amino acids, including all three branched chain amino acids (Supplementary Fig. 5B).

## Clenbuterol treatment increases basal- and sleeping metabolic rate

Next, we investigated if 14 days of clenbuterol treatment increased energy expenditure and altered substrate utilization. For this, we did not only measure basal metabolic rate, but also determined sleeping metabolic rate (SMR) using whole-room calorimetry, as it is known that SMR is the most sensitive measure to detect changes in energy needs. We found that sleeping- and basal metabolic rate increased by 10.2% and 5.6% ($p = 0.002$ and $p < 0.001$), respectively (Fig. 5A and Supplementary Fig. 2A). These changes in sleeping and basal energy expenditure were accompanied by modest increases in both fat and carbohydrate oxidation, albeit that these effects did not reach statistical significance (Fig. 5B–D and Supplementary Fig. 2B, C). In addition to these physiological data reflecting in vivo substrate metabolism, we

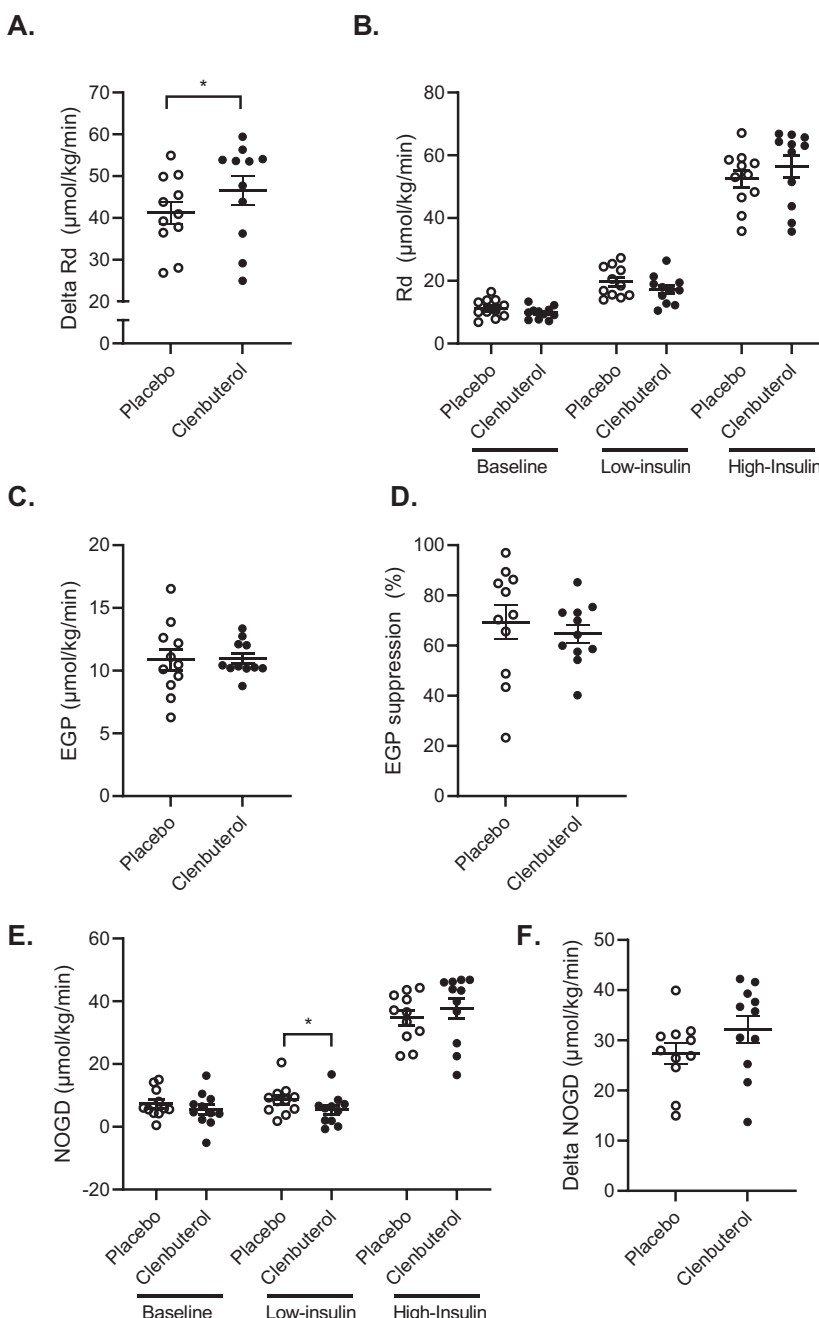

**Fig. 2 | Clenbuterol treatment enhances peripheral insulin-stimulated glucose disposal. A** Rate of glucose disposal (Rd) during the high-insulin phase corrected for baseline (µmol/kg/min) ($p = 0.032$). **B** Rate of glucose disposal during the hyperinsulinemic-euglycemic clamp at baseline, low- and high-insulin phases (µmol/kg/min). **C** Baseline endogenous glucose production (EGP) (µmol/kg/min). **D** Endogenous glucose production suppression during the low-insulin phase of the hyperinsulinemic-euglycemic clamp (%). **E** Non-oxidative glucose disposal (NOGD)

during the hyperinsulinemic-euglycemic clamp at baseline, low- and high-insulin phases (µmol/kg/min) (low-insulin: $p = 0.032$). **F** Non-oxidative glucose disposal during the high-insulin phase corrected for baseline (µmol/kg/min). All data were analyzed by means of a two-sided Wilcoxon signed-rank test. Placebo: $n = 11$, clenbuterol: $n = 11$. $*p < 0.05$. Data are presented as mean ± SEM. Source data are provided as a Source Data file.

also determined acylcarnitine profiles in plasma as a reflection of fat metabolism. In line with the indirect calorimetry data, fasting acyl-carnitine profiles remained largely unaffected upon clenbuterol treatment versus clenbuterol (Supplementary Fig. 6). Plasma glucose, insulin, and free fatty acids concentrations remained unaltered upon prolonged clenbuterol treatment (Fig. 6A, B, D), whereas plasma tri-glyceride concentrations were significantly lowered by 16.4% as com-pared to placebo ($p = 0.020$, Fig. 6C).

## Clenbuterol acutely increases energy expenditure and fat metabolism

Next to the effects of clenbuterol on blood pressure, heart rate, energy expenditure, and plasma substrates after 14 days of treatment, we also investigated the acute effects of clenbuterol administration. Thus, on day 1 of each treatment arm, blood pressure, heart rate, energy metabolism, and plasma substrates were determined over the course of 4 hours upon acute clenbuterol (20 µg) versus placebo

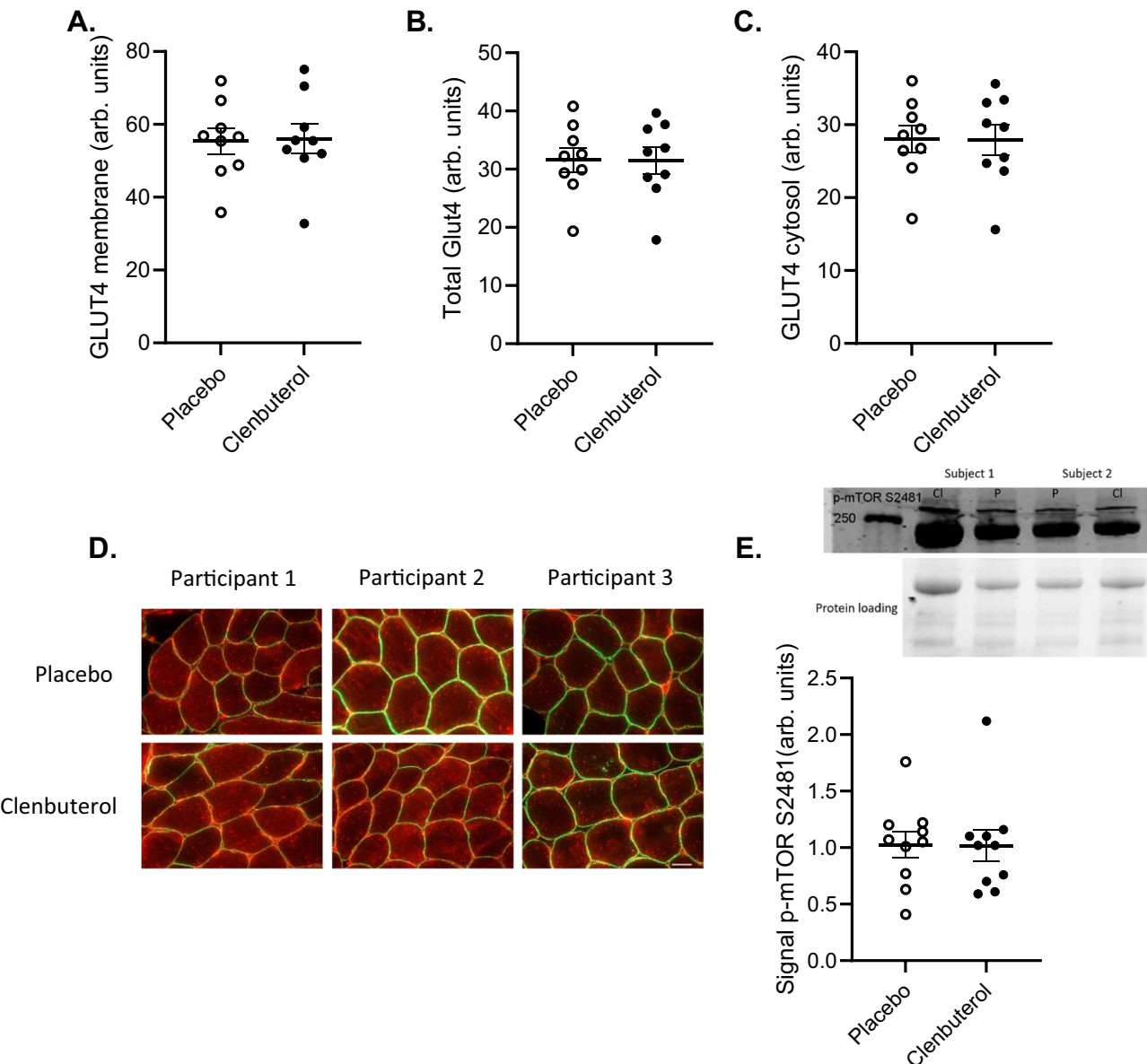

**Fig. 3 | Prolonged clenbuterol treatment does not affect skeletal muscle GLUT4 translocation or mTORC2 activation. A** GLUT4 content at the myocellular cell membrane (arb. units). **B** Total GLUT4 content (arb. units). **C** GLUT4 content in the myocellular cytosol (arb. units) **D** Representative images of GLUT4 immunohistochemistry staining for three participants, scale bar 50 μm. **E** Phosphorylation of mTORS2481 as a marker of mTORC2 activation (arb. units). Both samples of a participant were run on the same blot for comparison. All data were analyzed by means of a two-sided Wilcoxon signed-rank test. **A–D** $n = 9$ per group, **E** $n = 10$ per group. P = placebo, Cl = clenbuterol. Data are presented as mean ± SEM. Source data are provided as a Source Data file.

administration. Systolic- and diastolic blood pressure remained unaffected during the 4 hours following acute clenbuterol administration (20 μg) as compared to placebo (treatment effect: $p = 0.102$ and $p = 0.994$ for systolic and diastolic blood pressure, respectively, Fig. 7A, B). Heart rate showed a decrease over time in the placebo arm— an effect likely caused by the participants' prolonged supine position— which appeared to be blunted following clenbuterol administration. However, the treatment effect did not show statistical significance ($p = 0.207$, Fig. 7C). Similar to the effects of 14 days clenbuterol treatment, energy expenditure was significantly increased upon acute clenbuterol administration as compared to placebo (treatment effect: $p = 0.017$, Fig. 8A). In contrast, however, this increase was primarily attributed to an enhanced fat oxidation (treatment effect: $p = 0.017$, Fig. 8C), whereas carbohydrate oxidation was similar between

conditions (treatment effect: $p = 0.780$, Fig. 8B). The increased fat oxidation was accompanied by significantly elevated plasma FFA concentrations upon acute clenbuterol administration (treatment effect: $p = 0.005$, Fig. 8G), whereas plasma glucose and triglyceride concentrations remained unaltered (Figs. 8E, H). Plasma insulin levels were significantly different between the placebo and clenbuterol arms; however, this appeared mainly due to an unexpected difference in baseline insulin levels, which were determined prior to clenbuterol/ placebo intake (Fig. 8F).

## Discussion

Preclinical studies have shown that treatment with a selective $\beta_2$-AR agonist can improve skeletal muscle glucose uptake and whole-body glucose homeostasis in rodents, independent of changes in insulin

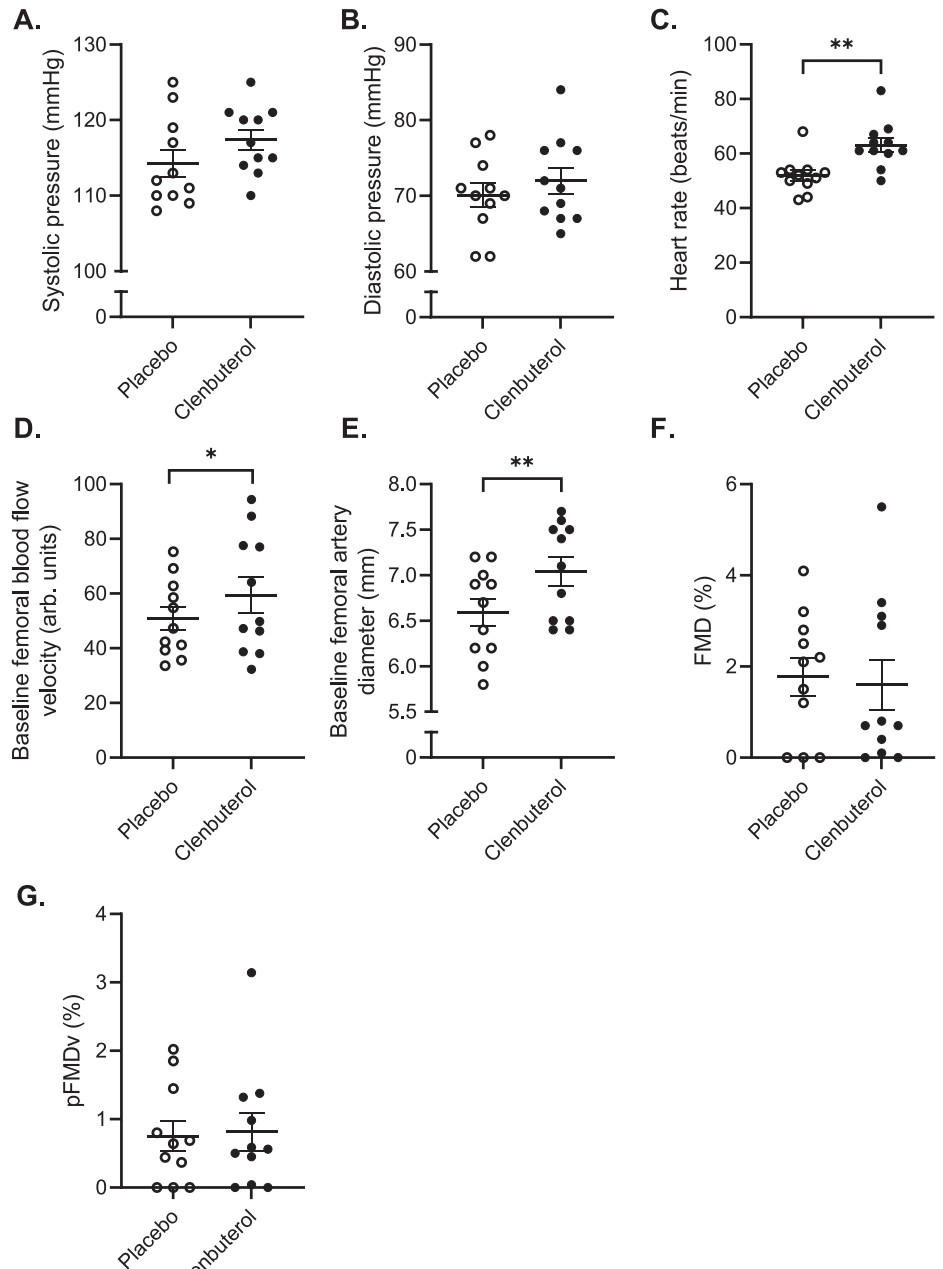

**Fig. 4 | 14 days clenbuterol treatment minimally affects blood pressure and heart rate but increases femoral artery blood flow velocity and diameters.** **A** Systolic pressure (mmHg). **B** Diastolic pressure (mmHg). **C** Heart rate (beats/min) ($p = 0.003$). **D** Baseline femoral artery blood flow velocity (arb. units) ($p = 0.014$). **E** Baseline femoral artery diameters (mm) ($p = 0.005$). **F** Flow-mediated vasodilation (%). **G** Flow-mediated vasodilation corrected for peak velocity flow stimulus (%). All data were analyzed by means of a two-sided Wilcoxon signed-rank test. Placebo: $n = 11$, Clenbuterol: $n = 11$. *$p < 0.05$, **$p < 0.01$. Data are presented as mean ± SEM. Source data are provided as a Source Data file.

concentrations[5–11]. However, the therapeutic potential of these data is largely unknown, as information on the effects of longer-term, systemic $\beta_2$-AR agonist treatment on skeletal muscle glucose uptake and whole-body metabolism in humans is limited. Therefore, we here investigated the effect of prolonged clenbuterol treatment (40 μg/day for 14 days) on peripheral insulin-stimulated glucose disposal in healthy young males, as assessed by means of a two-step hyperinsulinemic-euglycemic clamp. To gain further insights into the putative underlying mechanisms involved in $\beta_2$-AR agonist-mediated skeletal muscle glucose uptake, we used state-of-the-art techniques for detailed metabolic phenotyping of the study participants. We demonstrate that 14 days of clenbuterol treatment enhances

peripheral insulin-stimulated glucose disposal by ~13%, an effect primarily accounted for by an increased NOGD of ~18%. In addition, basal- and sleeping metabolic rates were significantly elevated upon clenbuterol treatment, whereas body weight- and composition remained unaltered. Lastly, clenbuterol treatment induced a marked increase in both femoral artery blood flow velocity and diameters whereas GLUT4 translocation and mTORC2 activation remained unaffected in overnight fasted muscle biopsies.

In the current study, an acute dose of clenbuterol (20 μg), as well as prolonged clenbuterol treatment, led to a significant increase in resting energy expenditure in comparison with placebo. Although the exact mechanisms through which selective $\beta_2$-AR agonists exert their

 

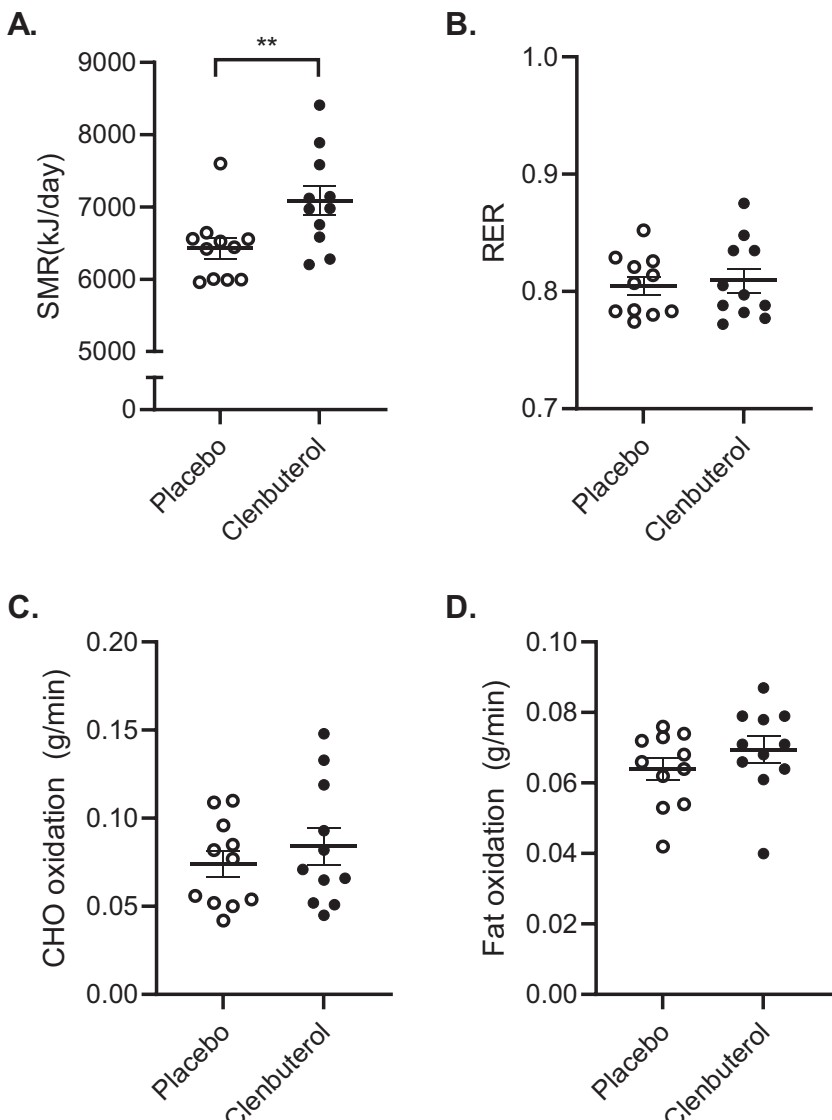

**Fig. 5 | Prolonged clenbuterol treatment enhances sleeping metabolic rate independent of selective increases in substrate oxidation. A** Sleeping metabolic rate (SMR) (kJ/min) ($p = 0.002$). **B** Respiratory exchange ratio. **C** Carbohydrate oxidation during the night (g/min). **D** Fat oxidation during the night (g/min). SMR sleeping metabolic rate, RER respiratory exchange ratio, CHO carbohydrate. All data were analyzed by means of a two-sided Wilcoxon signed-rank test. Clenbuterol: $n = 11$, Placebo: $n = 11$. **$p < 0.01$. Data are presented as mean ± SEM. Source data are provided as a Source Data file.

thermogenic effects remain elusive, it is likely that these effects are mediated via various energy-consuming processes in potentially skeletal muscle (i.e., through stimulation of the futile $Ca^{2+}$ cycle or $Na^+/K^+$-ATPase pumps[14]) or brown adipose tissue[15]. In line with previous studies[16–19], the increase in energy expenditure upon acute clenbuterol administration was primarily fuelled by enhanced fat oxidation and was accompanied by higher availability of plasma FFAs, likely mediated via $\beta_2$-AR-stimulated adipose tissue lipolysis. In contrast, the increased energy expenditure upon two weeks of clenbuterol treatment occurred independently of selective changes in fat oxidation or plasma FFA concentrations. In addition to these findings, plasma acylcarnitine concentrations were largely similar between clenbuterol and placebo treatments, thereby indicating that fat oxidation was not significantly altered.

Acute administration of $\beta_2$-AR agonists is also described as 'pro-diabetogenic' due to initial elevations in plasma glucose and insulin concentrations[6,20,21]. However, these detrimental effects of acute $\beta_2$-AR agonists administration appear to diminish over time, as longer-term

$\beta_2$-AR agonists treatment is not associated with increased plasma glucose and insulin concentrations[22–24]. In the current study, neither acute (20 µg) nor more prolonged clenbuterol administration significantly affected plasma glucose and insulin levels in healthy young males, likely due to the relatively low dose of clenbuterol applied. Two weeks of clenbuterol treatment did affect lipid metabolism, as fasting plasma triglyceride content was significantly reduced by 16.4%. These effects are presumably caused by an enhanced lipoprotein lipase (LPL) activity within skeletal muscle and/or brown adipose tissue, as previous rodent studies have demonstrated significant increases in LPL activity in these tissues upon clenbuterol or $\beta$-AR agonist treatment[25–27].

Previously, several in vitro and preclinical studies have highlighted the ability of $\beta_2$-AR agonists to enhance skeletal muscle glucose uptake and to improve glucose tolerance[3–11]. In line with these findings, two previous studies in humans[28,29] reported a ~2.0- and ~1.84-fold increase in leg glucose uptake at rest upon administration with the selective $\beta_2$-AR agonists salbutamol (24 mg orally) or terbutaline sulphate

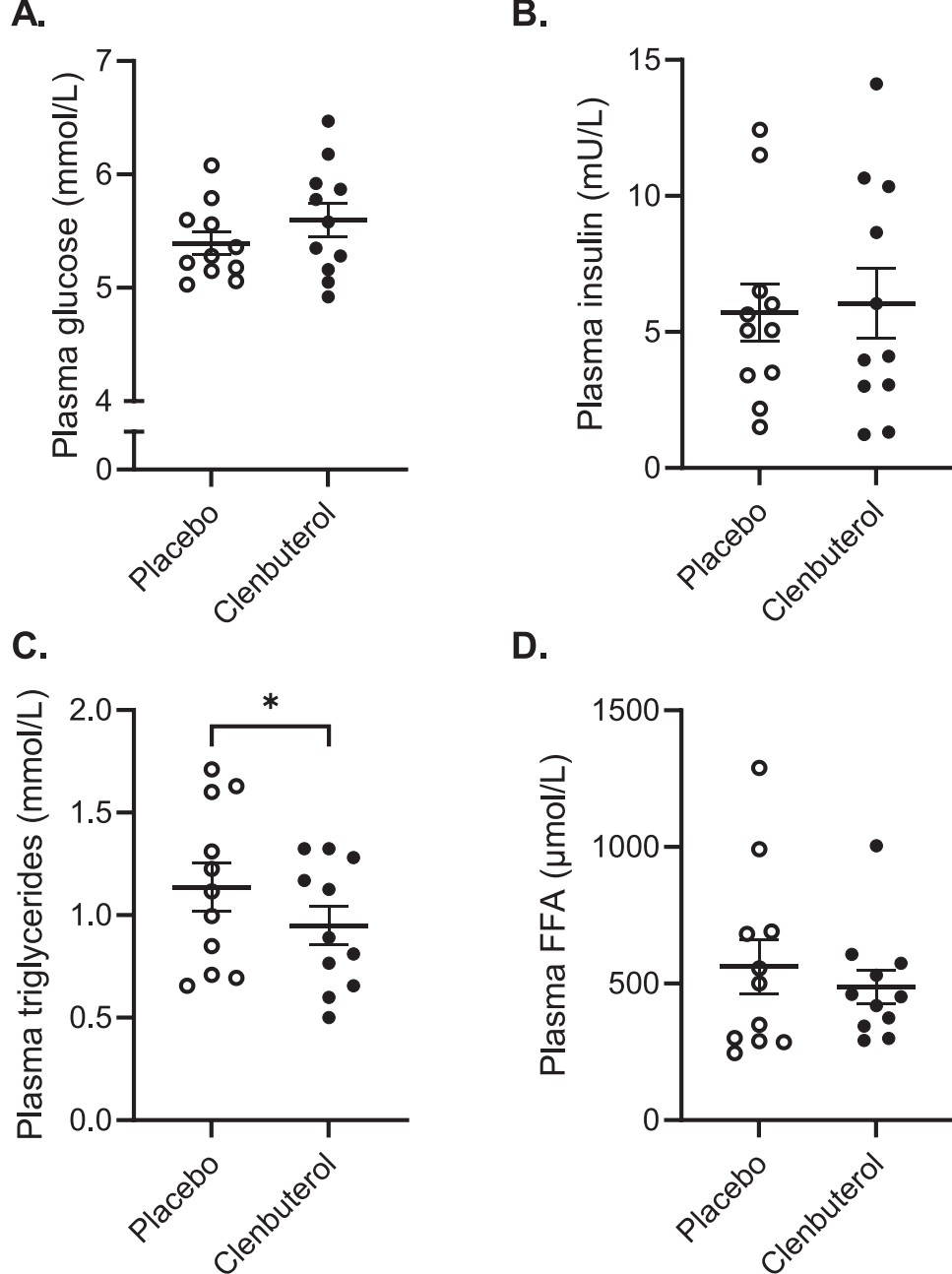

**Fig. 6 | The effect of prolonged clenbuterol treatment on fasting plasma substrate concentrations. A** Plasma glucose concentrations (mmol/L). **B** Plasma insulin concentrations (mU/L). **C** Plasma triglyceride concentrations (mmol/L) ($p = 0.02$). **D** Plasma free fatty acid concentrations (μmol/L). FFA free fatty acid. All data were analyzed by means of a two-sided Wilcoxon signed-rank test. Clenbuterol: $n = 11$, Placebo: $n = 11$. *$p < 0.05$. Data are presented as mean ± SEM. Source data are provided as a Source Data file.

(0.2–0.4 mg infusion), respectively. However, these studies involved an acute β$_2$-AR agonist stimulation, and, importantly, the observed effects were paralleled by significant increases in plasma insulin concentrations[28], thereby inherently affecting skeletal muscle glucose uptake. Information on the effects of more prolonged, systemic β$_2$-AR agonist stimulation on skeletal muscle glucose uptake and glucose homeostasis in humans is currently scarce. Next to the present study, merely one other study previously reported on prolonged (1–2 weeks), systemic treatment with a β$_2$-AR agonist (terbutaline sulphate; 5 mg, 3x/day orally) and showed increased glucose disposal during insulin infusion as well as insulin-stimulated NOGD in healthy males by ~29 and ~45%, respectively, independent of changes in insulin concentrations[22]. The beneficial effects of prolonged clenbuterol treatment on glucose

homeostasis observed in the present study are in line with the latter study and also occurred without differences in plasma insulin levels. Although direct comparison between these two studies is difficult due to the different β$_2$-AR agonist used, it is important to highlight that the beneficial effects of β$_2$-AR agonist treatment reported here were already achieved at lower therapeutic doses (40 μg/day orally, recommended dose clenbuterol: 40–80 μg/day), whereas Scheidegger et al[22]. used a relatively high dose of terbutaline sulphate (15 mg/day orally, recommended dose: 7.5–15 mg/day). Nevertheless, combined these studies clearly indicate the potential of β$_2$-AR agonists in stimulating human skeletal muscle glucose uptake.

Given that clenbuterol is infamous for its repartitioning effects upon high doses (i.e., increasing lean mass whilst reducing fat mass)[7–9,12],

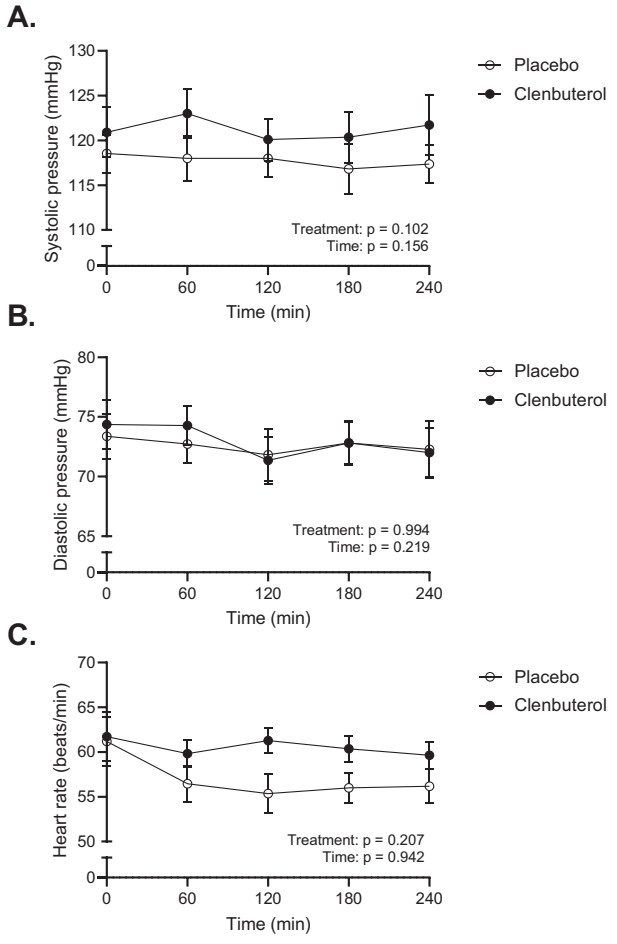

**Fig. 7 | Acute clenbuterol treatment (20 µg) does not affect blood pressure and heart rate. A** Systolic blood pressure (mmHg). **B** Diastolic blood pressure (mmHg). **C** Heart rate (beats/min). All data were analyzed by means of a linear mixed model analysis without correction for multiple comparisons. Clenbuterol: $n = 11$. Placebo: $n = 11$. Data are presented as mean ± SEM. Source data are provided as a Source Data file.

the increased insulin-stimulated peripheral glucose uptake could potentially be attributed to a higher skeletal muscle mass, as previously suggested[7,8]. Nevertheless, in this study, we were unable to attribute the effects of clenbuterol on glucose uptake to an increased lean mass, as body weight- and composition remained unaffected. In line with these findings, previous studies have reported similar beneficial effects of clenbuterol treatment on glucose homeostasis in rodents without changes in body composition[6,10,11], thereby further supporting the notion that β2-AR agonist treatment can improve glucose homeostasis independent of alterations in lean mass. To study more subtle changes in protein metabolism, we also determined mTORC1 phosphorylation, a marker of protein synthesis, as well as fasting levels of plasma amino acids. Interestingly, 2 weeks of clenbuterol treatment significantly lowered fasting plasma concentrations of 12 amino acids, including all three branched-chain amino acids (BCAAs). As mTORC1 activation—and thus protein synthesis—remained unaffected, it is likely that clenbuterol treatment reduces skeletal muscle protein degradation, potentially through an enhanced insulin action. However, the current study could not provide further data to directly support this notion. Nevertheless, it is important to highlight that elevated plasma concentrations of BCAAs in individuals with overweight/obesity and patients with T2DM correlate with insulin resistance[30–32] and pharmaceutical stimulation of BCAA oxidation via sodium phenylbutyrate treatment significantly increases

insulin sensitivity T2DM patients[33]. Based on these data, our observed improvements in insulin-stimulated glucose disposal may—at least in part—be attributable to changes in BCAA metabolism.

An alternative mechanism by which clenbuterol could affect skeletal muscle glucose uptake is through increased tissue perfusion. More specifically, selective β2-AR agonists are well-known for their ability to stimulate (peripheral) tissue blood flow[28,34,35], an important rate-limiting step in skeletal muscle glucose uptake[36]. Although femoral artery blood flow—here assessed by means of Doppler ultra-sonography—likely does not accurately reflect peripheral tissue capillary perfusion, it is tempting to hypothesize that our reported ~17% increase femoral artery blood flow is associated with enhanced capillary perfusion and thereby potentially glucose uptake. Although a previous study has been unable to establish an apparent link between acute β2-AR agonists administration, increases in blood flow, and skeletal muscle glucose uptake[28], it remains possible that improvements in glucose uptake upon prolonged β2-AR agonist treatment are —at least in part—mediated via enhanced tissue perfusion.

Although the exact underlying molecular mechanisms remain elusive, the improvements in glucose homeostasis upon selective β2-AR agonists in preclinical models appear to be largely dependent on mTORC2 signaling within the skeletal muscle[5,11]. Thus, a previous study uncovered a novel pathway stimulating GLUT4-mediated skeletal muscle glucose uptake via activation of the β2-AR independent of the insulin and AMPK pathways, namely via mTORC2[5]. In line with these findings, a recent study demonstrated that clenbuterol-mediated improvements in glucose homeostasis are reduced in mice ablated of skeletal muscle-specific Rictor, a key subunit of mTORC2[11]. Unbiased transcriptomics and metabolomics performed in the latter study further highlighted the central role of mTORC2 in clenbuterol-induced improvements in glucose homeostasis[11]. To investigate if mTORC2 signaling and GLUT4 translocation were involved in the increase in β2-AR agonist-mediated glucose uptake that we observed, we here measured skeletal muscle GLUT4 translocation as well as the phosphorylation of mTOR S2481, a marker for the activation of mTORC2, in muscle biopsies collected in the overnight fasted state. However, we were unable to detect differences in both GLUT4 trans-location as well as mTORC2 activation, upon prolonged clenbuterol treatment.

Several limitations can be identified in our study. Firstly, only male participants were included in our study. It is therefore important that future studies within this field are performed with both men and women to investigate if similar effects occur in both sexes. Secondly, we did not measure urinary nitrogen to accurately assess whole-body protein oxidation, but for the calculation of substrate metabolism instead assumed protein oxidation to be a fixed percentage of energy expenditure. However, this assumption might not hold true in situations of enhanced energy expenditure (i.e., clenbuterol treatment), during which variable protein oxidation can occur. Important differences in substrate oxidation could therefore be missed. Thirdly, we recognize that the sample size in this study was limited, which could potentially explain the lack of findings with respect to GLUT4 translocation or mTORC2 activation. Lastly, our muscle biopsies were taken in the overnight fasted, unsti-mulated state, whereas glucose uptake was elevated upon insulin sti-mulation. In other words, analysis of insulin-stimulated skeletal muscle material upon β2-AR agonist treatment may be imperative to address the mechanisms underlying the observed improvement in peripheral insulin sensitivity.

In conclusion, we here demonstrate that a two-week treatment with the selective β2-AR agonist clenbuterol enhances insulin sensi-tivity, mainly via increased insulin-stimulated NOGD, in healthy young males. These beneficial effects were accompanied by increases in sleeping metabolic rate, improvements in plasma triglyceride con-centrations, reductions in plasma amino acid concentrations, and increases in arterial blood flow velocity. However, skeletal muscle

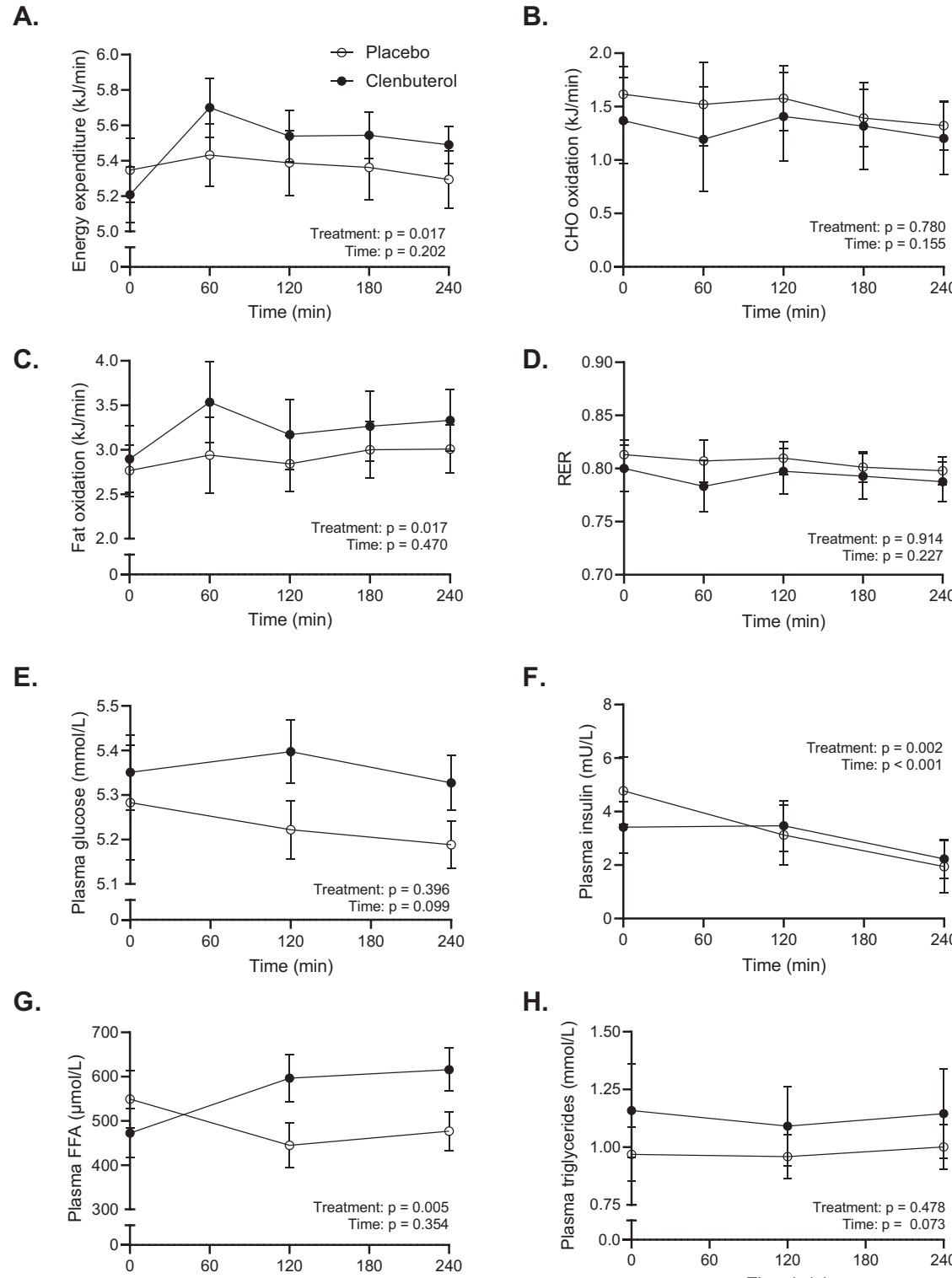

**Fig. 8 | Acute clenbuterol administration (20 μg) enhances energy expenditure and plasma free fatty acid concentrations. A** Energy expenditure (kJ/min). B. Carbohydrate oxidation (CHO) (kJ/min). C. Fat oxidation (kJ/min). D. Respiratory exchange ratio. E. Plasma glucose concentrations (mmol/L). F. Plasma insulin concentrations (mU/L). G. Plasma free fatty acid (FFA) concentrations (μmol/L). H. Plasma triglyceride concentrations (mmol/L). $N = 10$ for indirect calorimetry data and n = 11 for plasma analyses. FFA = free fatty acids. All data were analysed by means of a linear mixed model analysis without correction for multiple comparisons. Data are presented as mean ± SEM. Source data are provided as a Source Data file.

mTORC2 activation and GLUT4 translocation, assessed in biopsies taken in the overnight fasted state, remained unaffected upon two-week clenbuterol treatment. Given these effects in young, healthy volunteers, it is tempting to postulate that prolonged treatment with a selective β2-AR agonist also has the potential to have a marked beneficial effect on insulin sensitivity in volunteers with disturbed glucose homeostasis, such as in T2DM patients. However, given the well-known (cardiovascular) side effects associated with prolonged systemic β2-AR

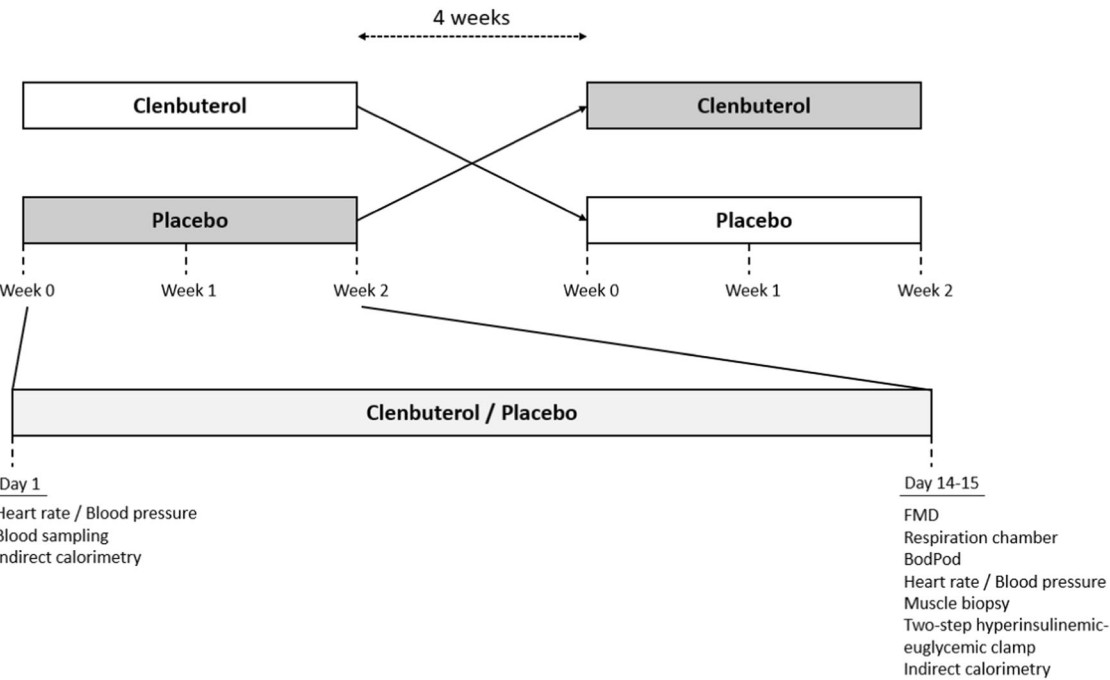

**Fig. 9 | Overview study design.** FMD flow-mediated vasodilation.

agonist treatment, the development of novel, highly selective β₂-AR agonists is crucial for further translation into the clinic. Otherwise, further exploration of the underlying (molecular) mechanisms involved could lead to the identification of viable therapeutic targets and the development of a novel class of therapeutic compounds for the treatment of T2DM.

## Methods

### Ethical approval
Data was collected at Maastricht University, Maastricht, The Netherlands between the 13th of August 2019 and the 23rd of April 2021. The study was temporarily halted from the 16th of March until the 1st of July 2020 due to the national COVID-19 lockdown in The Netherlands. In total, 12 participants were enrolled and randomized in the study, with 1 participant dropping out due to the COVID-19 pandemic (see Fig. 1 for a CONSORT inclusion flow chart). The study was performed in accordance with the declaration of Helsinki and was reviewed and approved by the Ethics Committee of the Maastricht University Medical Centre + (NL67646.068.18). The study was registered at ClinicalTrials.gov under the identifier NCT03800290. All participants provided their written informed consent prior to screening. The trial reporting complies with the International Committee of Medical Journal Editors Guidelines.

### Participants
Healthy, young, male participants were included in the study. Participants were of Caucasian origin, aged between 18 and 30 years with a BMI between 20–25 kg/m², and did not participate in organized or structured physical exercise. Participants were recruited in Maastricht and its direct surroundings by means of flyers and advertisements on the internet. All participants were screened for eligibility and were excluded if one of the following conditions were met: cardiovascular diseases (as determined by ECG, blood pressure measurements, and medical questionnaires); respiratory diseases; unstable body weight (weight gain or loss >5 kg in the last three months); intention to lose or gain body weight (through caloric restriction or exercise); excessive alcohol and/or drug abuse; hypokalaemia; hyperthyroidism; anaemia; epilepsy; smoking; renal and/or

liver insufficiency; medication use is known to hamper the participant's safety during the study procedures.

### Experimental design
Based on a paired samples T-test power calculation, an expected standard deviation of 9.7 μmol/kg/min[37], and an expected mean difference of 25%, we calculated that 11 participants would be required to complete the study in order to reject the null hypothesis with a power of 80% and a type 1 error probability of 0.05. In a randomised, placebo-controlled, double-blinded, cross-over design participants received either the selective β₂-AR agonist clenbuterol hydrochloride (2 × 20 μg/day) (Spiropent, Hikma Pharmaceuticals, Portugal) or a placebo for two weeks with a four week wash-out period (Fig. 9). Due to the national COVID-19 lockdown in the Netherlands, two participants had an extended wash-out of 25 weeks each. Participants were randomly allocated to a study arm by means of controlled randomisation and the allocation sequence was generated in blocks of four by an independent researcher. Clenbuterol hydrochloride and placebo capsules were prepared by Radboud University Pharmacy, Nijmegen, The Netherlands. For this, clenbuterol hydrochloride tablets (20 μg) were encapsulated and filled with the inactive compound lactose. Clenbuterol hydrochloride capsules contained 20 μg clenbuterol hydrochloride/capsule. Placebo capsules only contained the inactive compound lactose. Containers were sequentially numbered to conceal the allocation sequence. Participants were instructed to consume two capsules of clenbuterol (2 × 20 μg = 40 μg/day) or placebo daily with breakfast and dinner. All unused capsules were returned by the participant. During the study period, participants were asked to maintain their normal eating and physical activity habits. Prior to all visits, participants were asked to refrain from alcohol and physical activity other than their normal daily routine for 72 hours.

**Acute effects of clenbuterol on heart rate, blood pressure, energy expenditure, and plasma substrates.** At the start of each study arm (day 1), participants arrived at the research unit in an overnight fasted state (i.e., no food consumption after 22:00 the night before). Body weight, heart rate, and blood pressure were measured followed by the

collection of a fasting blood sample. Subsequently, baseline energy expenditure was measured for 30 minutes using indirect calorimetry ($T = -30$ till $T = 0$). Afterwards, participants consumed their first capsule of the study arm (i.e., 20 μg of clenbuterol hydrochloride or placebo) under supervision of the researcher ($T = 0$). The following 4 hours after initial drug intake, blood samples were collected every two hours ($T = 120$ and $240$), whereas heart rate and blood pressure were measured every hour ($T = 60, 120, 180$, and $240$). In addition, energy expenditure and substrate oxidation were measured for 30 minutes every hour ($T = 60$–$90, 120$–$150, 180$–$210$, and $240$–$270$).

**Half-way safety check-up.** After one week of treatment (day 8), a half-way safety check-up was performed. During this visit, participants were asked about potential emerged side-effects and their overall well-being.

**Effects of two-week clenbuterol treatment on glucose homeostasis.** Following two weeks of treatment (day 14), participants arrived at the research unit at 17:00 to first determine endothelial function and blood flow responses using flow-mediated vasodilation of the femoral artery in the leg. Afterwards (-18:00), participants received a standardized meal during which the last capsule of the respective study period was consumed. Participants then stayed overnight in a respiration chamber to determine sleeping metabolic rate and substrate oxidation. The following morning, body composition was determined by means of air displacement plethysmography (BodPod). Afterwards, blood pressure and heart rate were measured, and a muscle biopsy was taken in the overnight fasted state. Subsequently, a two-step hyperinsulinemic-euglycemic clamp with indirect calorimetry was performed for the determination of hepatic and peripheral insulin-stimulated glucose disposal.

## Primary and secondary research outcomes

The primary research outcome of this study was peripheral insulin sensitivity upon 2 weeks of clenbuterol versus placebo treatment, expressed as the change in insulin-stimulated glucose disposal in the high-insulin phase over baseline (ΔRd) during the two-step hyperinsulinemic-euglycemic clamp. The secondary research outcome was skeletal muscle GLUT4 translocation, as assessed by means of immunohistochemistry and confocal microscopy.

## Exploratory research outcomes

Exploratory outcomes of this research included blood pressure, basal femoral artery blood flow, and diameter, flow-mediated dilation, (sleeping) energy expenditure and substrate oxidation, plasma substrate concentrations, body composition, and protein expression in skeletal muscle.

## Two-step hyperinsulinemic-euglycemic clamp

A two-step hyperinsulinemic-euglycemic clamp with primed continuous co-infusion of D-[6,6-2H2]-glucose (0.04 mg/kg/min) was performed to assess hepatic and whole-body insulin sensitivity[38]. Suppression of endogenous glucose production (EGP) during low-insulin infusion (10 mU/m$^2$/min) was used as a marker for hepatic insulin sensitivity. Peripheral glucose disposal (Rd) was measured during low- and high-insulin infusion (10 mU/m$^2$/min and 40 mU/m$^2$/min, respectively). Glucose appearance (Ra) and glucose disposal (Rd) were calculated according to Steele's single pool non-steady state equations[39]. Delta Rd (ΔRd) was determined in the low- and high-insulin phases by calculating the Rd insulin$_{low/high}$ minus Rd basal. The volume of distribution of glucose was assumed to be 0.160 L/kg. Energy expenditure and substrate oxidation were measured over a 30-minute period with indirect calorimetry during baseline, low-insulin, and high-insulin phases of the clamp. NOGD was defined as Rd minus carbohydrate oxidation. Delta NOGD (ΔNOGD) in the low- and high insulin phases was determined by

calculating NOGD insulin$_{low/high}$ minus NOGD basal. The percentual suppression of plasma FFAs during the low- and high-insulin phases of the clamp as compared to fasting plasma FFA values was used as a marker of adipocyte insulin sensitivity.

## Skeletal muscle biopsy and immunohistochemistry

On day 15 of each study period, a skeletal muscle biopsy was obtained from the *m.* vastus lateralis under local anaesthesia (1% lidocaine without adrenaline) according to the Bergström technique[40]. Part of the muscle biopsy was embedded into Tissue-Tek and frozen in melting isopentane for immunohistochemical analysis. The remainder was immediately frozen in melting isopentane. For one participant, the muscle biopsy could not be obtained due to technical reasons. Muscle tissue from another participant was of inadequate quality for immuno-histochemical analyses, resulting in a sample size of $n = 9$. All samples were stored at −80 °C until further analyses. For GLUT4 imaging, immunofluorescence assays were performed on 5 μm thick sections, cut from the skeletal muscle biopsies. After fixation with acetone, sections were incubated with a polyclonal rabbit antibody directed to GLUT4 (1:50; ab33780; Abcam, Abcam B.V., Amsterdam, The Netherlands) and a mouse monoclonal antibody directed to caveolin (1:25; 610421; BD Biosciences, Vianen, The Netherlands), and the appropriate Alexa Fluor 555 (1:500; A21428, Invitrogen)–and Alexa Fluor 488 (1:200; A21121, Invitrogen) -conjugated secondary antibodies. Images were acquired on a Nikon E800 fluorescence microscope (Nikon Europe BV, Amsterdam, the Netherlands) coupled to Nikon DS-Fi1c colour CCD camera (Nikon) using NIS-Elements imaging software (Nikon). Images were captured with identical exposure time and gain settings in paired ("before-and-after") samples. Without any adjustments with respect to color intensity, brightness, or contrast, 8-bit images were quantified using ImageJ (NIH, Bethesda, USA) (Schneider et al. 2012). Cell membranes were thresholded and selected. Based on the thresholded cell membranes the muscle fibers were selected and were shrunken with 12 pixels to have a clean separation of the cytosol and the cell membrane. Subsequently, the mean intensity of GLUT4 fluorescence was measured on the cell membranes and in the cytosol.

## Indirect calorimetry

An automated respiratory gas analyser and ventilated hood system (Omnical, IDEE, Maastricht, The Netherlands) were used to measure whole-body oxygen consumption and carbon dioxide production of participants in a supine position over a 30-minute period. For the acute effects of clenbuterol, indirect calorimetry data for one participant was excluded due to a technical error. Total energy expenditure and carbohydrate- and fat oxidation was calculated according to Brouwers' equation[41], and protein oxidation was set at 12.4% of basal energy requirements as determined by Weir's equation[42]. For the determination of SMR and substrate oxidation, participants spent the night from day 14 to 15 in a respiration chamber (Omnical, IDEE, Maastricht, The Netherlands). SMR was defined as the lowest energy expenditure for 3-consecutive hours as calculated according to Weir's equation (42). Total energy expenditure and substrate oxidation during the night were calculated as described above.

## Heart rate and blood pressure

Prior to the measurement of heart rate and blood pressure, participants were rested for 10 minutes in a supine position. Heart rate and blood pressure were measured three times in a row by means of an automatic inflatable cuff (Omron Healthcare, Hamburg, Germany) on days 1 and 15.

## Femoral artery flow-mediated vasodilation

On day 14, femoral artery flow-mediated vasodilation was assessed by Doppler ultrasonography (MyLab™-Gamma, Esaote, Maastricht, The

Netherlands) by using a 13–4 MHz linear transducer. Measurements were performed in B-mode with Doppler to assess continuous artery diameter and blood flow velocity profiles. After recording a 3-minute baseline reference period, a pneumatic cuff placed around the participant's right leg was inflated at 200 mmHg for 5 minutes. Following this hypoxic period, the cuff was released and images were recorded for another 5 minutes. The echo images were analyzed offline to determine diameter and velocity profiles over the entire measurement period with a custom-written MatLab program (MyFMD 2015, professor A.P.G. Hoeks, Department of Biomedical Engineering, Maastricht University, Maastricht, the Netherlands). The FMD response was quantified as the maximal percentage change in post-occlusion arterial diameter relative to baseline diameter. The FMD corrected for peak velocity was calculated by dividing the FMD by the percentage change in post-occlusion peak velocity flow.

### Body composition
Body weight- and composition were determined at -06:00 on day 15 following an overnight fast (from -18:00 the previous day) using air displacement plethysmography (BodPod®, COSMED, Inc., Rome, Italy).

### Plasma substrate analyses
Blood samples were collected in EDTA, NaF, and serum tubes. After collection, EDTA and NaF tubes were stored on ice and spun down at 4 °C for 10 minutes at 1300 RCF. Serum tubes were stored at room temperature for at least 30 minutes to allow coagulation and were thereafter spun down at 21 °C for 10 minutes at 1300 RCF. Blood plasma and serum were collected and stored at −80 °C until further analyses. Plasma glucose (Horiba, Montpellier, France), free fatty acids (WAKO, Neuss, Germany), and triglycerides (Sigma, St Louis, USA) concentrations were measured by means of a colorimetric analysis using a Cobas Pentra C400 analyzer (Horiba, Montpellier, France). Plasma insulin concentrations were measured by means of an ELISA (Crystal Chem, Elk Grove Village, USA).

### Plasma metabolomics
Plasma acylcarnitines and amino acids were determined by flow injection MS/MS and LC-MS/MS, respectively, as described previously[43,44].

### Western blot
Analyses of protein expression in skeletal muscle biopsies ($n = 10$) were performed by means of Western blotting. Muscle tissue (10–15 mg) was homogenised in Bio-Plex Cell Lysis buffer (Bio-Rad Laboratories, Hercules, CA, USA). Stain-free gradient (4–15%) TGX gels (Bio-Rad Laboratories) or Invitrogen Bolt 4–12% Bis-Tris Protein gels (ThermoFisher Scientific, Waltham, Massachusetts, USA) were loaded with equal quantities of protein (5 or 10 µg/well) and after electrophoresis transferred to a nitrocellulose membrane (Trans-Blot Turbo Transfer System, Bio-Rad laboratories, Hercules, CA, USA). Blots were incubated overnight with the primary p-mTORC2 S2481 (1:1000, #2974, Cell Signalling, Danvers, MA, USA), p-mTORC1 S2448 (1:1000, #2971, Cell Signalling, Danvers, MA, USA), VDAC (1:1000, sc-390996, Santa Cruz Biotechnology, Dallas, Texas, USA), TOMM20 (1:10,000, ab186734, Abcam, Cambridge, UK), or an OxPhos antibody cocktail (1:1000, ab110411, Abcam, Cambridge, UK). The following morning, blots were incubated with the appropriate IRDye800-conjugated secondary antibody (Donkey anti-rabbit IRDye800; 1:10,000; 926-32213; Li-COR, or Donkey anti-mouse IRDye800; 1:10,000; 626-32212; Li-COR, Lincoln, Nebraska, USA). Proteins were visualized and quantified by means of a CLx Odyssey Near Infrared Imager (LI-COR, Lincoln, Nebraska, USA). Images of uncropped western blots can be found in the Supplementary Information and the Source Data file.

### Statistical analyses
Data were statistically analyzed with SPSS IBM version 27 and graphs were created in GraphPad Prism version 9.3.1. The primary and secondary outcomes were analyzed by means of a non-parametric Wilcoxon signed-rank test. Explorative outcomes were analysed by means of a non-parametric Wilcoxon signed-rank test in the case of paired data or by linear mixed model analyses for data acquired over time. Linear mixed model analyses with random intercept were performed to analyze changes over time as compared to baseline upon clenbuterol versus placebo administration. For this, time points were normalized for baseline values and a linear mixed model analyses was performed on the timepoints $T = 60, 120, 180$, and $240$. Period, time, treatment, and the time*treatment interaction term were used as fixed factors, whereas participant was used as random factor. If the interaction term was not statistically significant, it was omitted from the model and only the main effects (i.e., treatment and time) were reported. An unstructured covariance structure was used for these analyses. Statistical analyses were considered significant if $p < 0.05$. All data are presented as mean ± SEM unless stated otherwise.

### Reporting summary
Further information on research design is available in the Nature Portfolio Reporting Summary linked to this article.

## Data availability
The study protocol can be found in the supplementary information (Supplementary Note 1). For academic purposes, the de-identified and processed participant data can be requested from the corresponding author (J.hoeks@maastrichtuniversity.nl) with no end date, following the completion of a signed data access agreement form. De-identified data will be shared due to participant privacy. Source data are provided in this paper.

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

## Acknowledgements

This study was supported by a grant from ZonMW and the Dutch Diabetes Research Foundation (J.H.) and the Nutrim NWO graduate program (S.v.B.). Funding agencies had no influence on the study design, data collection and analysis, and manuscript writing. The authors would like to thank Jorg Sander and Martin Vervaart for their assistance during this study.

## Author contributions

S.v.B., T.B., M.H., B.H., P.S., and J.H. were involved in designing the study. Experiments were performed by S.v.b., Y.B., F.V. C.F., N.C., G.S.,

E.K., F.V., J.J., E.S., P.J., and A.G. Data were analyzed by S.v.B, F.V., G.S., E.K., J.J., E.S., P.J., A.G., R.H., P.S., and J.H. S.v.B., B.H., P.S., and J.H. drafted the manuscript. The manuscript was reviewed and edited by R.H., B.H., P.S., and J.H. All authors read and approved the final version of the manuscript before submission. J.H. is the guarantor of the study.

## Competing interests

T.B. owns stock in Atrogi AB. The remaining authors declare no competing interests.

## Additional information

Sten M. M. van Beek[1], Yvonne M. H. Bruls[2], Froukje Vanweert [1], Ciarán E. Fealy[1], Niels J. Connell [1], Gert Schaart [1], Esther Moonen-Kornips[1], Johanna A. Jörgensen[1], Frédéric M. Vaz [3,4,5], Ellen T. H. C. Smeets[1], Peter J. Joris[1], Anne Gemmink [1], Riekelt H. Houtkooper[3,4,6], Matthijs K. C. Hesselink [1], Tore Bengtsson [7], Bas Havekes[1,8], Patrick Schrauwen [1] & Joris Hoeks [1]✉

[1]Department of Nutrition and Movement Sciences, NUTRIM School of Nutrition and Translational Research in Metabolism, Maastricht University, Maastricht, the Netherlands. [2]Department of Radiology and Nuclear Medicine, NUTRIM School of Nutrition and Translational Research in Metabolism, Maastricht University Medical Center+, Maastricht, the Netherlands. [3]Laboratory Genetic Metabolic Diseases, Amsterdam UMC location University of Amsterdam, Meibergdreef 9, Amsterdam, Netherlands. [4]Amsterdam Gastroenterology, Endocrinology, and Metabolism, Amsterdam, The Netherlands. [5]Core Facility Metabolomics, Amsterdam UMC location University of Amsterdam, Amsterdam, The Netherlands. [6]Amsterdam Cardiovascular Sciences, Amsterdam, The Netherlands. [7]Department of Molecular Biosciences, The Wenner-Gren Institute, Stockholm University, Stockholm, Sweden. [8]Department of Internal Medicine, Division of Endocrinology and Metabolic Disease, NUTRIM School of Nutrition and Translational Research in Metabolism, Maastricht University Medical Center, Maastricht, The Netherlands. ✉e-mail: J.Hoeks@maastrichtuniversity.nl

