## [Peer Review File · Nature Communications]

Effect of β 2-agonist treatment on insulin-stimulated peripheral glucose disposal in healthy men in a randomised placebo-controlled trialREVIEWER COMMENTS

Reviewer #1 (Remarks to the Author):

Comments to Authors

In this manuscript, van Beek and co-workers investigated the effect of β 2-adrenergic agonist clenbuterol effect on skeletal muscle glucose uptake and whole-body fuel metabolism. They report that β 2-adrenergic agonist enhances insulin stimulated glucose uptake and non-oxidative glucose disposal, increased basal and sleeping metabolic rates, femoral blood flow velocity, and decreased fasting plasma triglycerides. However, they could not find any change in GLUT-4 translocation and mTORC2 activation. They propose that increased metabolic rate is largely due to increased fatty acid oxidation based on indirect calorimetry-based measurement.

Although β 2-adrenergic receptor agonist was studied in the late 1980s and 1990s these studies were not pursued largely because the lack of increase in lean body mass shown in animal models. It is important to see whether these fuel metabolism changes are associated with any evidence of skeletal muscle protein anabolism or catabolism. Since these studies were done only for two weeks, it is unrealistic to expect any changes in body composition based on air displacement approach. Although increased metabolic rate and fuel oxidation may offer potential benefits, increased oxygen consumption and fuel oxidation does not necessarily mean that ATP production increased and there was no increase in oxidative stress.

Although, this manuscript reports some interesting physiological measurements the impact of these findings will be enhanced by new insights based on additional measurements.

Specific Points:

- 1) Some of the changes occurred are of statistically borderline significance which as authors indicate, may be related to small sample size. This is especially striking for fat oxidation. The authors did not measure urinary nitrogen in the fasted state (during indirect calorimetry) for protein oxidation but assumed to be fixed which is usually fine but in situation with higher metabolic rate variable protein oxidation can occur. Thus important differences in protein oxidation that impacts of non-protein RQ may be missed.
- 2) It would be useful to measure mitochondrial enzyme activities including citrate synthase and ECT pathways enzymes such as cytochrome-c- oxidase to determine whether mitochondrial capacity is enhanced
- 3) More insight on the fuel metabolism in muscle and plasma will substantially enhance the impact of this manuscript. It would be useful to measure plasma and muscle amino metabolites and perhaps acylcarnitine which may give greater insight on the fuel metabolism pattern in response to β 2 agonist.
- 4) In fasting state when metabolic rate is high, it is important to exclude any protein catabolism, however, net protein changes are unlikely to occur in two weeks' time and therefore body composition change won't help. It is important but expensive undertaking, to perform RNA sequencing of the muscle that may identify some pathways involved in protein synthesis and degradation and determine which pathways are activated by administration of clenbuterol.

5) The authors speculated that high FFA may have stimulated increased metabolic rate but evidence based on studies done many especially M.D. Jensen's group indicates that RMR drives lipolysis and increased FFA and its oxidation

Reviewer #2 (Remarks to the Author):

This study investigated the effects of a two week treatment with the selective β 2-agonist clenbuterol on insulin-stimulated skeletal muscle glucose uptake in eleven healthy young males in a randomised, placebo-controlled, double39 blinded, cross-over study. Clenbuterol treatment improved insulin-stimulated glucose uptake and non-oxidative glucose disposal assessed by hyperinsulinemic-euglycemic clamp, without changes in body mass- and composition. Clenbuterol increased sleeping- and basal metabolic rate, as well as femoral artery blood flow velocity, whereas fasting plasma triglyceride concentrations decreased.

GLUT4 translocation and mTORC2 activation were unaffected by clenbuterol treatment in overnight fasted muscle biopsies. These results highlight the potential of β 2-agonist treatment in improving skeletal muscle glucose uptake and underscore the therapeutic value of this pathway for the treatment of type 2 diabetes.

Comments:

The manuscript is well written and the idear is original and interesting. However I have a few major comments that need to be addressed:

Given the well-known (cardiovascular) side-effects associated with prolonged systemic β 2-AR agonist treatment, further exploration on the underlying (molecular) mechanisms involved is needed to identify viable therapeutic targets.

This needs to be added to the abstract as well.

Furthermore, the discussion of the clinical application of the drug with acceptable safety profile needs to be discussed.

The sample size is very limited (11) and elaborations towards power calculations should be included and discussed.

Other comments:

What are the noteworthy results:

Clenbuterol treatment improved insulin-stimulated glucose uptake and non-oxidative glucose disposal assessed by hyperinsulinemic-euglycemic clamp, without changes in body mass- and composition.

Will the work be of significance to the field and related fields:

Please include a discussion/elaboration of the findings in contrast to treatments that also have profound

effects on inulin sensitivity and reduction of glucose levels. See eg.
Healthy weight loss maintenance with exercise, liraglutide, or both combined
JR Lundgren, C Janus, SBK Jensen, CR Juhl, LM Olsen, RM Christensen, ...
New England Journal of Medicine 384 (18), 1719-1730

There was no effect on reduction in body weight - in contrast to the treatments mentioned above.
Would this be a limitation given that weight loss has profound effects on insulin sensitivity (see above)?

How does it compare to the established literature: The study and design is original.

Does the work support the conclusions and claims, or is additional evidence needed: See comments above.

Reviewer #3 (Remarks to the Author):

line 2: It would be helpful to mention the study design in the title (if possible).

line 329: It would be useful to briefly mention in a sentence the primary and secondary outcomes assessed.

line 331: Number of participants is not needed in the methods section. It is presented in the results.

line 332: It would be good to mention the race ("Caucasian") in the inclusion criteria as it is mentioned in the trial registration.

line 352: Is there a predefined allocation ratio (eg 1:1)? Is a carry over effect possible in this type of intervention?

line 717 (Supplemental Figure 1):

- 4 people in total did not meet inclusion criteria or met exclusion criteria? Wording is not very clear
- 3 withdrawals: Technically they had not been assigned to a study arm at that moment, so the term withdrawal is not very accurate. Maybe rephrase to "declined to participate"?

Additional comments:

1. It would be important to briefly describe the sample size calculation and the parameters that

determined it (ie calculation method, clinical important difference, power, a-level , dropout rate).

2. Which randomisation method was used (eg simple randomisation)?

3. Statistical analysis (line 502):

a) Given the small sample size have you considered tests of normality and non-parametric tests?

b) Results of linear mixed models are mentioned mainly as p-values over time?

c) It would be useful to specify the formula used to measure the % change in the outcomes. The linear mixed models calculate effect estimates rather than percentage changes.

Reviewer comments

Reviewer #1 (Remarks to the Author):

Comments to Authors

In this manuscript, van Beek and co-workers investigated the effect of β 2-adrenergic agonist clenbuterol effect on skeletal muscle glucose uptake and whole-body fuel metabolism. They report that β 2-adrenergic agonist enhances insulin stimulated glucose uptake and non-oxidative glucose disposal, increased basal and sleeping metabolic rates, femoral blood flow velocity, and decreased fasting plasma triglycerides. However, they could not find any change in GLUT-4 translocation and mTORC2 activation. They propose that increased metabolic rate is largely due to increased fatty acid oxidation based on indirect calorimetry-based measurement.

Although β 2-adrenergic receptor agonist was studied in the late 1980s and 1990s these studies were not pursued largely because the lack of increase in lean body mass shown in animal models. It is important to see whether these fuel metabolism changes are associated with any evidence of skeletal muscle protein anabolism or catabolism. Since these studies were done only for two weeks, it is unrealistic to expect any changes in body composition based on air displacement approach.

Although increased metabolic rate and fuel oxidation may offer potential benefits, increased oxygen consumption and fuel oxidation does not necessarily mean that ATP production increased and there was no increase in oxidative stress.

Although, this manuscript reports some interesting physiological measurements the impact of these findings will be enhanced by new insights based on additional measurements.

Specific Points:

1) Some of the changes occurred are of statistically borderline significance which as authors indicate, may be related to small sample size. This is especially striking for fat oxidation. The authors did not measure urinary nitrogen in the fasted state (during indirect calorimetry) for protein oxidation but assumed to be fixed which is usually fine but in situation with higher metabolic rate variable protein oxidation can occur. Thus important differences in protein oxidation that impacts of non-protein RQ may be missed.

We agree with the reviewer that we cannot exclude changes in protein oxidation in light of the higher metabolic rate that we observed upon treatment with the β 2 agonist, which may have affected the calculations for carbohydrate and fat oxidation. We have added this point to the limitation section of the discussion of the manuscript.

2) It would be useful to measure mitochondrial enzyme activities including citrate synthase and ECT pathways enzymes such as cytochrome-c- oxidase to determine whether mitochondrial capacity is enhanced

Thank you for this suggestion. Since we did not have sufficient muscle tissue left for enzymatic activity assays, we used our existing protein lysates to measure protein expression of VDAC, TOMM20, and structural components of the OxPhos complexes by means of Western Blotting, as markers for mitochondrial capacity. The results of these measurements have been added to the results section and supplementary figures. In short, these new data demonstrate that 2-weeks of clenbuterol treatment did not affect protein expression of OxPhos, VDAC, and TOMM20, thereby indicating that mitochondrial capacity was unaffected by the intervention.

3) More insight on the fuel metabolism in muscle and plasma will substantially enhance the impact of this manuscript. It would be useful to measure plasma and muscle amino metabolites and perhaps acylcarnitine which may give greater insight on the fuel metabolism pattern in response to β 2 agonist.

As mentioned above, we did not have sufficient muscle tissue left to perform additional analyses in muscle. In response to the reviewer's comment however, we have now analysed plasma amino acid as well as acylcarnitine profiles by means of targeted metabolomics in overnight fasted plasma samples after 2 weeks of clenbuterol vs. placebo, to gain more insight in the fuel metabolism. In short, we found that clenbuterol treatment significantly reduced fasting plasma concentrations of 12 amino acids, possibly due to an enhanced amino acid uptake by skeletal muscle. Plasma acylcarnitine profiles on the other hand, remained largely unaffected by clenbuterol treatment, indicating that there were no large changes in fat metabolism, at least not reflected in plasma. These new data were added to the result section of the manuscript and are also briefly discussed in the discussion section.

4) In fasting state when metabolic rate is high, it is important to exclude any protein catabolism, however, net protein changes are unlikely to occur in two weeks' time and therefore body composition change won't help. It is important but expensive undertaking, to perform RNA sequencing of the muscle that may identify some pathways involved in protein synthesis and degradation and determine which pathways are activated by administration of clenbuterol.

Thank you for this comment. Although it would be interesting to perform RNA sequencing in order to explore pathways putatively involved in protein synthesis and -degradation upon clenbuterol treatment, the lack of available muscle tissue and indeed also the costs involved, prevented us from performing this analysis. In order to investigate potential effects of clenbuterol treatment on muscle protein synthesis/breakdown, we therefore measured the activation of mTORC1 (as a marker of protein synthesis) by Western Blotting and, as mentioned above, determined amino acid profiles in the plasma. In short, we demonstrate that the activation of mTORC1 was not significantly affected upon clenbuterol treatment, whereas the concentration of 12 plasma amino acids was markedly reduced. These results have been added to the results section of the paper and are also discussed in the discussion section.

5) The authors speculated that high FFA may have stimulated increased metabolic rate but evidence based on studies done many especially M.D. Jensen's group indicates that RMR drives lipolysis and increased FFA and its oxidation.

There may have been some misunderstanding in relation to this point, for which we apologize. We agree with the reviewer that it is unlikely that high FFAs stimulated the observed increased in resting metabolic rate and that this was not clearly described in the manuscript. Therefore, we have now rewritten this part of the discussion.

Reviewer #2 (Remarks to the Author):

This study investigated the effects of a two week treatment with the selective β_2 -agonist clenbuterol on insulin-stimulated skeletal muscle glucose uptake in eleven healthy young males in a randomised, placebo-controlled, double-blind, cross-over study. Clenbuterol treatment improved insulin-stimulated glucose uptake and non-oxidative glucose disposal assessed by hyperinsulinemic-euglycemic clamp, without changes in body mass- and composition. Clenbuterol increased sleeping- and basal metabolic rate, as well as femoral artery blood flow velocity, whereas fasting plasma triglyceride concentrations decreased.

GLUT4 translocation and mTORC2 activation were unaffected by clenbuterol treatment in overnight fasted muscle biopsies. These results highlight the potential of β_2 -agonist treatment in improving skeletal muscle glucose uptake and underscore the therapeutic value of this pathway for the treatment of type 2 diabetes.

Comments:

The manuscript is well written and the idea is original and interesting. However I have a few major comments that need to be addressed:

Given the well-known (cardiovascular) side-effects associated with prolonged systemic β_2 -AR agonist treatment, further exploration on the underlying (molecular) mechanisms involved is needed to identify viable therapeutic targets.

This needs to be added to the abstract as well.

Thank you. We have now added the following sentence to the abstract: "However, given the well-known (cardiovascular) side-effects of systemic β_2 -agonist treatment, further exploration on the underlying mechanisms is needed to identify viable therapeutic targets".

Furthermore, the discussion of the clinical application of the drug with acceptable safety profile needs to be discussed.

We would like to emphasize that this study was not performed to assess the clinical applicability of clenbuterol to improve glucose homeostasis in T2DM patients, as the side-effects of clenbuterol clearly prevent this. More specifically, this study was performed as a proof-of-principle study to investigate if treatment with a selective β_2 -agonist could improve skeletal muscle glucose uptake; hence providing a starting point for the potential development of a novel class of anti-diabetic drugs. These novel drugs would then have to be specific β_2 -agonists with a low affinity for β_1 - and β_3 -adrenergic receptors, to minimize side-effects. We now further emphasize this point of view in the conclusion section of the paper.

The sample size is very limited (11) and elaborations towards power calculations should be included and discussed.

In line with the comment of the reviewer, we have included the power calculation in the methods section under the section "Ethical Approval".

Other comments:

What are the noteworthy results:

Clenbuterol treatment improved insulin-stimulated glucose uptake and non-oxidative glucose disposal assessed by hyperinsulinemic-euglycemic clamp, without changes in body mass- and composition.

Will the work be of significance to the field and related fields:

Please include a discussion/elaboration of the findings in contrast to treatments that also have profound effects on insulin sensitivity and reduction of glucose levels. See eg. Healthy weight loss maintenance with exercise, liraglutide, or both combined JR Lundgren, C Janus, SBK Jensen, CR Juhl, LM Olsen, RM Christensen, ... New England Journal of Medicine 384 (18), 1719-1730.

It is important to realize that our findings of improved peripheral insulin-stimulated glucose uptake upon 2 weeks of clenbuterol treatment were obtained in young, lean and completely healthy individuals. Obviously, this also hampers direct comparison of our findings with more common insulin sensitizing interventions that are performed in older, overweight/obese patient or risk groups. Follow-up research would be required to study the effectiveness of stimulating β_2 -AR to improve insulin sensitivity in people with or at risk for developing type 2 diabetes.

There was no effect on reduction in body weight - in contrast to the treatments mentioned above. Would this be a limitation given that weight loss has profound effects on insulin sensitivity (see above)?

We agree with the reviewer that common dietary lifestyle interventions that induce weight loss are extremely effective in treating type 2 diabetes. In the current study however, we could demonstrate that β_2 -agonist treatment significantly improves insulin sensitivity in young, healthy males, independent of weight loss. We believe this is a highly relevant finding that could eventually contribute to a positive disease outcome for T2DM patients (once the right drugs have been developed and side-effects can be circumvented). In this context, one can draw comparisons with exercise training, also considered as one of the best treatment options for type 2 diabetes, which also improves peripheral insulin sensitivity irrespective of weight loss.

How does it compare to the established literature: The study and design is original.

Does the work support the conclusions and claims, or is additional evidence needed: See comments above.

Reviewer #3 (Remarks to the Author):

line 2: It would be helpful to mention the study design in the title (if possible).

We have added the study design in the title, which is now as follows: “ β_2 -agonist treatment promotes insulin-stimulated skeletal muscle glucose uptake in healthy males in a randomised placebo-controlled trial”

line 329: It would be useful to briefly mention in a sentence the primary and secondary outcomes assessed.

According to Nature Communications, and CONSORT, guidelines, we have added a section in the Methods entitled “Primary and secondary research outcomes”. In this section, both the primary and secondary research outcomes are now specified.

line 331: Number of participants is not needed in the methods section. It is presented in the results.

We have removed the number of participants in line 331.

line 332: It would be good to mention the race ("Caucasian") in the inclusion criteria as it is mentioned in the trial registration.

In line with the comment of the reviewer, we have added “Caucasian” to the inclusion criteria in line 332.

line 352: Is there a predefined allocation ratio (eg 1:1)? Is a carry-over effect possible in this type of intervention?

During the study, we tried to have an equal allocation ratio between the two interventions arms. Upon inclusion, subjects were randomly allocated to a study arm by means of controlled randomisation and the allocation sequence was generated in blocks of four by an independent researcher. However, due to several subjects withdrawing from participating prior to the start of the intervention period, the final allocation ratio was 7:4 (i.e. 7 subjects starting with clenbuterol, and 4 subjects starting with placebo).

In cross-over studies, carry-over effects are a possibility. To minimize the risk for carry-over effects, we deliberately choose a wash-out period of 4 weeks. Given the half-life of clenbuterol (35h), 97% of the investigational product was removed from the body after 7 days following withdrawal of the drug. To also test for potential order effects during the study, we calculated the change in rate of glucose disposal between the clenbuterol and placebo arms ($\Delta R_{d_{clenbuterol}} - \Delta R_{d_{placebo}}$) and statistically analysed whether this change was significant different between the allocation sequences. Although the sample size is limited, we did not observe any significant differences between the two sequences, indicating that an order effect did not occur.

line 717 (Supplemental Figure 1):

- 4 people in total did not meet inclusion criteria or met exclusion criteria? Wording is not very clear

In line with the author guidelines of Nature Communications, we have replaced the previous inclusion flow chart with the CONSORT inclusion flow chart. As such, this issue raised by this reviewer has been resolved.

- 3 withdrawals: Technically they had not been assigned to a study arm at that moment, so the term withdrawal is not very accurate. Maybe rephrase to “declined to participate”?

As mentioned above, we have replaced the previous inclusion flow chart with the CONSORT inclusion flow chart, which has resolved the issue raised by the reviewer.

Additional comments:

1. It would be important to briefly describe the sample size calculation and the parameters that determined it (ie calculation method, clinical important difference, power, a-level , dropout rate).

Additional information with respect to the sample size calculation (and the parameters that determined it) has been added to the Methods in the section “Ethical Approval”.

2. Which randomisation method was used (eg simple randomisation)?

During the study, we made use of controlled randomisation to allocate subjects to a treatment arm. This information has been added to the Methods in the section “Experimental design”.

3. Statistical analysis (line 502):

a) Given the small sample size have you considered tests of normality and non-parametric tests?

Thank you for your question. The reviewer may have missed that we indeed assessed the paired data for normality by means of a Shapiro-Wilk normality test. In case data was normally distributed, a two-sided Paired Student’s T-test was used, whereas non-normally distributed was analysed by means of a Wilcoxon Signed-Rank test.

b) Results of linear mixed models are mentioned mainly as p-values over time?

In the linear mixed model analyses, Time, Treatment, and the Time*Treatment interaction terms were used as fixed factors. If the interaction term was not statistically significant, it was omitted from the model and only the main effects (i.e. treatment and time) were reported with their respective p-values. Since we were unable to demonstrate significant interaction effects, only time and treatment effect p-values were reported.

c) It would be useful to specify the formula used to measure the % change in the outcomes. The linear mixed models calculate effect estimates rather than percentage changes.

For the linear mixed model analyses, we analysed changes over time as compared to baseline upon clenbuterol versus placebo administration. Thus, in the experiments regarding the acute effects of clenbuterol administration, individual baseline values were subtracted from the other time points of the same subject. Afterwards, a linear mixed model analyses was performed on the time points T = 60, 120, 180, and 240. As such, we did not calculate the percentage change to perform the linear mixed model analyses. This has been clarified in the statistical analyses section of the Methods.

REVIEWER COMMENTS

Reviewer #1 (Remarks to the Author):

It is intriguing that clenbuterol treatment significantly reduced fasting plasma concentrations of 12 amino acids but the speculation that "it occurred possibly due to an enhanced amino acid uptake by skeletal muscle" is unlikely especially no activation of mTOR signaling was noted. This reviewer is unaware of any human studies where AAs decrease due to increased muscle protein synthesis. The more likely reason is reduced muscle protein degradation that may occur with enhanced insulin action. Unfortunately no supporting data is available. It is important to briefly discuss this limitation

Reviewer #4 (Remarks to the Author):

1. Sample size calculation:

If the primary outcome is "change in Rd", the sample size calculation should use statistics of change in Rd. Is 9.7 μ mol/kg/min the standard deviation of delta Rd? I cannot find the number in reference 37. Mean difference of 25% is not clear either.

This description can be moved to experimental design section.

2. Statistical power:

The study does not have sufficient power as the recruitment goal is not achieved. The primary finding is inconclusive since we can not know whether the non-significance of "change in Rd" is due to limited sample size.

The normality test is affected by the limited sample size too.

3. Primary and secondary outcome

List all secondary outcomes you analyzed. Some were not mentioned.

4. Line 441-443 should be moved to results.

5. Statistical method:

As mentioned above, the normality test may not have enough power and is likely to show non-significance. So nonparametric method should be considered.

Are sequence effect and/or carry over effect tested in the mixed model?

Clarify whether $p < 0.05$ is considered significance? If so, do not show p values > 0.05 (i.e. $p = 0.08$) in the figures. For secondary outcomes, p values may need to be adjusted for multiple comparisons.

Clearly specify methods for primary and secondary outcomes. Present primary analyses (for primary outcome) first, followed by secondary analysis.

6. Results

Show results for primary outcome first (e.g. delta Rd), followed by secondary outcomes.

For example, Fig 2 should show Rd instead of bp and hr.

The author's responses to Reviewer 3's other questions are acceptable.

Reviewer comments

Reviewer #1 (Remarks to the Author):

It is intriguing that clenbuterol treatment significantly reduced fasting plasma concentrations of 12 amino acids but the speculation that "it occurred possibly due to an enhanced amino acid uptake by skeletal muscle" is unlikely especially no activation of mTOR signaling was noted. This reviewer is unaware of any human studies where AAs decrease due to increased muscle protein synthesis. The more likely reason is reduced muscle protein degradation that may occur with enhanced insulin action. Unfortunately no supporting data is available. It is important to briefly discuss this limitation

We agree with the reviewer that our hypothesis regarding the fact that the reduction 12 plasma amino acids points towards enhanced skeletal muscle amino acid uptake upon clenbuterol versus placebo treatment, was too speculative. We have now changed this in the discussion and state that a reduced muscle protein degradation (possibly due to an enhanced insulin action) is a more likely explanation. We also added that no data was acquired during this study to directly support this notion, as a limitation.

Reviewer #4 (Remarks to the Author):

1) Sample size calculation:

If the primary outcome is "change in Rd", the sample size calculation should use statistics of change in Rd. Is $9.7 \mu\text{mol/kg/min}$ the standard deviation of delta Rd? I cannot find the number in reference 37. Mean difference of 25% is not clear either. This description can be moved to experimental design section.

The primary outcome is defined as the difference in insulin-stimulated glucose disposal (rate of disappearance, Rd) between clenbuterol treatment vs. placebo (which we anticipated to be 25%). The Rd is determined during an hyperinsulinemic-euglycemic clamp, both in the basal state (prior to starting the infusion of insulin) and during the insulin-stimulated state. The change in Rd or delta Rd (i.e. Rd upon insulin infusion minus Rd at baseline) then reflects the insulin-stimulated glucose disposal. To be honest, we and others use these outcome markers (Rd during insulin vs delta Rd) inconsistently, also because often the basal Rd is not affected by most interventions, and therefore both are a good reflection of peripheral insulin sensitivity. Also here, basal Rd is not affected. Indeed, here the original power calculation was erroneously based on statistics of Rd, rather than delta Rd. Fortunately, due to years of experience and optimization of the internal procedures, the variation in the clamp data in the current study was smaller than in the study we used for the original power calculation (ref 37), and therefore the observed difference between clenbuterol and placebo was statistically significant. Nonetheless, due to the small sample size (as also highlighted in the reviewer's second point), we have now analyzed all data in the manuscript using the non-parametric Wilcoxon signed rank test instead of the paired samples T-test.

2) Statistical power:

The study does not have sufficient power as the recruitment goal is not achieved. The primary finding is inconclusive since we can not know whether the non-significance of "change in Rd" is due to limited sample size. The normality test is affected by the limited sample size too.

Due to the limited sample size, which indeed affects the normality test, we now amended the statistical analyses to the non-parametric Wilcoxon signed rank test throughout the entire paper. The main conclusions of the manuscript were not affected by this change.

3) Primary and secondary outcome

List all secondary outcomes you analyzed. Some were not mentioned.

In the current study, GLUT4 translocation was the only secondary outcome parameter. All other parameters that were assessed during this study were of an explorative nature.

4) Line 441-443 should be moved to results.

Lines 441-443 were moved to the results section.

5) Statistical method:

As mentioned above, the normality test may not have enough power and is likely to show non-significance. So a nonparametric method should be considered.

We agree with the comment of the reviewer and have now analyzed all outcomes using the non-parametric Wilcoxon signed rank test. The main conclusions of the manuscript were not affected by this change.

Are sequence effect and/or carry over effect tested in the mixed model?

Sequence effects and carry-over were not tested in the previous linear mixed model analyses. To account for potential sequence effects, we now added a new fixed variable in the linear mixed model named 'period' to test for sequence effects and amended the manuscript accordingly. Importantly, the addition of this fixed factor did not change the outcomes.

To test for potential carry-over effects influencing the acute measurements, we calculated the change in baseline energy expenditure and plasma glucose concentrations between the clenbuterol and placebo arms and statistically analysed whether this change was different between the allocation sequences. For these analyses, we specifically chose to compare baseline energy expenditure and plasma glucose concentrations since these variables were significantly affected by prolonged (14 days) clenbuterol treatment. Although the sample size is limited, we did not observe any significant differences between the two sequences, indicating that an order effect did not occur.

Clarify whether $p < 0.05$ is considered significance? If so, do not show p values > 0.05 (i.e. $p = 0.08$) in the figures. For secondary outcomes, p values may need to be adjusted for multiple comparisons.

Thank you, all p values > 0.05 were removed from the figures. However, as GLUT4 translocation was the only secondary outcome during this study, no multiple comparisons were performed.

Clearly specify methods for primary and secondary outcomes. Present primary analyses (for primary outcome) first, followed by secondary analysis.

The manuscript has been revised accordingly.

6) Results

Show results for primary outcome first (e.g. ΔR_d), followed by secondary outcomes. For example, Fig 2 should show R_d instead of bp and hr .

Thank you, the manuscript has been revised accordingly.

The author's responses to Reviewer 3's other questions are acceptable.